# Constraints on the martian crust away from the InSight landing site

Jiaqi Li [1] ✉, Caroline Beghein[1], Scott M. McLennan [2], Anna C. Horleston [3], Constantinos Charalambous[4], Quancheng Huang[5], Géraldine Zenhäusern [6], Ebru Bozdağ[5], W. T. Pike[4], Matthew Golombek[7], Vedran Lekić[8], Philippe Lognonné [9] & W. Bruce Banerdt [7]

The most distant marsquake recorded so far by the InSight seismometer occurred at an epicentral distance of $146.3 \pm 6.9°$, close to the western end of Valles Marineris. On the seismogram of this event, we have identified seismic wave precursors, i.e., underside reflections off a subsurface discontinuity halfway between the marsquake and the instrument, which directly constrain the crustal structure away (about 4100−4500 km) from the InSight landing site. Here we show that the Martian crust at the bounce point between the lander and the marsquake is characterized by a discontinuity at about 20 km depth, similar to the second (deeper) intra-crustal interface seen beneath the InSight landing site. We propose that this 20-km interface, first discovered beneath the lander, is not a local geological structure but likely a regional or global feature, and is consistent with a transition from porous to non-porous Martian crustal materials.

After the successful touchdown of the InSight lander[1], the subsequent detection of marsquakes[2], and the initial inference of the shallow crustal structure[3], a family of 94 low-frequency marsquakes[4,5] have been detected and recorded by the Seismic Experiment for Interior Structure (SEIS) seismometer[6]. In the first two and a half years, all the marsquakes with known epicentral distances occurred within a distance of fewer than 100 degrees[7,8], and most of those with known back azimuths clustered at epicentral distances around 30 degrees (i.e., about 1700 km) to the east of the InSight landing site[9,10], near Cerberus Fossae (Fig. 1a).

Various seismological methods have been applied to the InSight seismic data to constrain the crustal structure. For example, using receiver functions, Knapmeyer-Endrun et al. (2021) (ref. 11) obtained two sets of possible crustal models beneath the InSight landing site: a two-layer model with a Moho depth of $20 \pm 5$ km and a three-layer model with a crustal thickness of $39 \pm 8$ km (though with a weaker wave-speed contrast across this possible deeper Moho). Subsequent studies from Kim et al. (2021) (ref. 12) and Durán et al. (2022) (ref. 13) based on more comprehensive datasets and different methods confirmed the existence of two seismic discontinuities within the crust (i.e., at -8 km and ~ 20 km, respectively) and favored the existence of the third discontinuity at its base (i.e., at ~40 km) thus indicating a three-layer crust. Results from ambient noise auto-correlation[14,15] have also revealed a discontinuity at about 21 km depth, consistent with the base of the second layer of the receiver function studies. In addition, by combining receiver function results and SH-wave reflection observations, Li et al. (2022) (ref. 16) found the presence of seismic anisotropy in the shallowest 8 km of the crust at the landing site.

Although the methods employed above were effective at providing the first seismic model of the Martian crust, all the seismic body-wave studies reflect the structure beneath the InSight landing site. Constraining the crustal structure at other locations is, however, crucial to fully understand the formation and evolution of the crust, and to improve global models of crustal thickness based on gravity data[17].

[1]Department of Earth, Planetary, and Space Sciences, University of California, Los Angeles, CA 90095, USA. [2]Department of Geosciences, Stony Brook University, Stony Brook, NY 11794–2100, USA. [3]School of Earth Sciences, University of Bristol, Bristol, UK. [4]Department of Electrical and Electronic Engineering, Imperial College London, London, UK. [5]Department of Geophysics, Colorado School of Mines, Golden, CO, USA. [6]Institute of Geophysics, ETH Zurich, Zurich, Switzerland. [7]Jet Propulsion Laboratory, California Institute of Technology, Pasadena, CA 91109, USA. [8]Department of Geology, University of Maryland, College Park, MD, USA. [9]Université Paris Cité, Institut de physique du globe de Paris, CNRS, Paris F-75005, France. ✉e-mail: jli@epss.ucla.edu

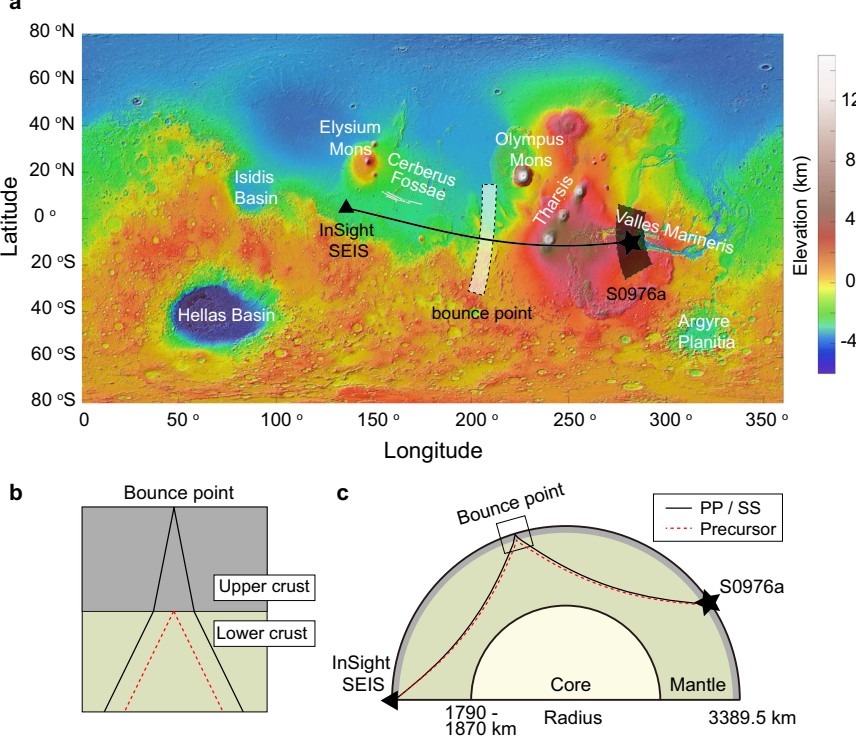

**Fig. 1 | Locations of the marsquake, seismic station, and bounce point. a** Global topographic map from the Mars Orbiter Laser Altimeter (MOLA[33]). The triangle marks the location of the InSight landing site, the star shows one of the possible locations of the marsquake (see Methods section), and the ray path is in black. The uncertainty of the marsquake location from seismological constraints is indicated by the shaded area (in black). The shaded region (in white) between the event and the station marks the possible corresponding locations of the bounce point. **b** Structure beneath the bounce point of the SS (or PP) waves (the box region in (**c**) assuming one layer in the upper crust. The dashed red lines are the ray paths for the SS (or PP) precursor reflected off the intra-crustal interface. The solid black lines are the ray paths for the SS (or PP) phase with a free-surface reflection. **c** Ray paths for the SS (or PP, in solid black) and its precursor (in dashed red). The triangle and star mark the InSight SEIS seismometer and event S0976a, respectively.

Although surface waves have been recently detected for the first time from two meteorite impacts[18,19] and the largest marsquake ever recorded[20], surface waves constrain the average structures along the event-lander path[19,21,22] and are less sensitive to subsurface discontinuities. The present work aims at providing seismic constraints on the crustal structure at a location far from the InSight lander.

In this study, we made use of the most distant marsquake yet recorded, event S0976a, which occurred on the 976th InSight Mars solar day, or sol (i.e., August 25th, 2021). Its epicentral distance is estimated to be $146.3 \pm 6.9°$ and its back azimuth is $101 \pm 25°$ (ref. 7). The epicenter is located near the western end of Valles Marineris (shaded area in Fig. 1a), the largest canyon system in the solar system[23] to the east of the Tharsis region.

At such a large epicentral distance, the direct P-wave (compressional wave) and S-wave (shear wave) are blocked by the Martian core with a radius estimated to be about 1,790-1,870 km[13,24–26]. However, the PP and SS phases, which bounce off the free surface of the planet halfway between the quake and the instrument, can be identified on the seismograms (Fig. 2a, b).

## Results and discussion
### InSight seismic observations
The PP phase arrived 1013 s after the origin time[7] (i.e., 2021-08-25 03:32:20) on the vertical component (Fig. 2a and Supplementary Fig. 1). There is an early-arriving signal $10.2 \pm 0.2$ s before the PP phase (Fig. 2a). We interpret this signal to be the crustal PP precursor reflected off the crustal interface beneath the bounce point (Fig. 1b, c). This is because, like the PP phase, this precursor is only clearly observed on the vertical component and does not show up on the

tangential component (Supplementary Fig. 2a), and because of the precursor's waveform similarity to the PP phase (i.e., a peak in the autocorrelation coefficients in Fig. 2b). The amplitude ratio between the PP precursor and the PP phase is about $43 \pm 8\%$ (Supplementary Fig. 3), which is also comparable to the synthetic amplitude ratio (~25–40%) predicted by the receiver-function-derived crustal models beneath the InSight landing site (Supplementary Fig. 4).

The SS phase arrived 1856 s after the origin time on the tangential component (Fig. 2a and Supplementary Fig. 1). With a polarization filtering technique[27], which enhances the linearly polarized signals (see Methods section), most wave trains arriving before the SS phase are attenuated except for one signal at about $19.3 \pm 0.5$ s relative to the SS phase (Fig. 2b). Based on four lines of reasoning, we interpret this early-arriving signal to be the crustal SS precursor: First, the amplitude of this possible SS precursor is not attenuated after polarization filtering, indicating a linearly polarized arrival on the tangential component. Second, there are similarities between this potential SS precursor and the SS phase particularly after polarization filtering (i.e., a peak in the auto-correlation coefficients in Fig. 2b). Third, at the arrivals of the SS phase and this SS precursor, strong and coherent signals are only observed on the tangential component, not on the vertical component (Supplementary Fig. 2b). Last, the amplitude ratio between the SS precursor and the SS phase is about $40 \pm 16\%$ (Supplementary Fig. 3), which is comparable to the amplitude ratio (~25–40%) inferred from the landing site models (Supplementary Figs. 4 and 5).

Importantly, the timings of the crustal PP and SS precursors are consistent, i.e., differential arrival times of $10.2 \pm 0.2$ s for the PP-waves and $19.3 \pm 0.5$ s for the SS-waves yield a $V_P$-to-$V_S$ ratio of 1.81 to 1.98,

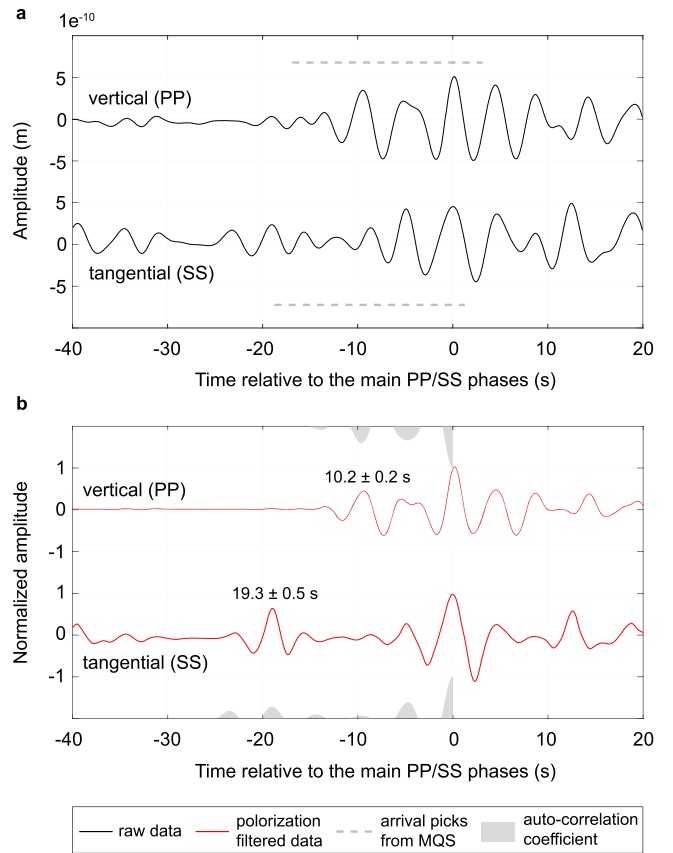

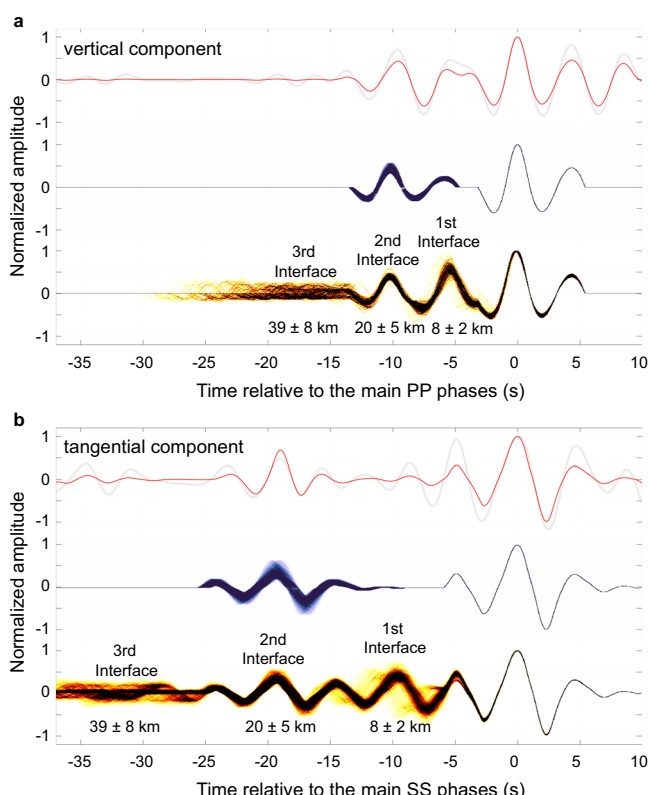

**Fig. 2 | InSight seismic observations of the PP and SS waves. a** The top and bottom traces are the displacement waveforms on the vertical component near the arrival of the PP phase and the tangential component near the arrival of the SS phase, respectively. Both traces are bandpass filtered between 2.8 s and 6 s with a two-pass second-order Butterworth filter. The signals at 0 s are the preferred PP and SS phases picked in this study, and the dashed grey bars indicate the PP and SS arrivals (with an uncertainty of 10 s) provided by Horleston et al. (2022) (ref. [7]). **b** Similar to (**a**) but with an additional polarization filtering[27] to enhance the body-wave signals. The differential arrivals of the PP/SS precursors are also annotated (see Methods section for the uncertainty estimations). The positive parts of the auto-correlation coefficients for the vertical and tangential components are shown as shaded waveforms on the top (multiplied by −1 to fit into the plot) and bottom axes, respectively.

**Fig. 3 | Seismic observations and synthetic waveform fits. a** The trace on the top panel (in red) is the PP wave train on the vertical component after polarization filtering, and the raw displacement record is also shown in light grey. The waveforms on the middle panel are the synthetic waveforms using the inverted models at the bounce point (shown in Fig. 4c). The seismograms on the bottom panel are synthetics using the receiver-function-derived models at the InSight landing site[11] (see Methods section for the synthetic waveform calculation). **b** Similar to (**a**), but for the SS wave train on the tangential component.

which overlaps with the range of 1.7–1.9 derived with receiver functions beneath the lander[11] (also shown in Supplementary Fig. 6).

## Forward modeling using landing site models

The SS (or PP) phase and its precursor share almost the same ray paths except for the regions near the bounce point (Fig. 1c); thus, the differential arrival time and amplitude ratio mainly reflect the structure beneath the bounce point.

In Fig. 3, we first used the receiver-function-derived models beneath the landing site[11] (also shown in Fig. 4a) to calculate the synthetic arrival time and amplitude of the PP and SS precursors using ray theory (see Methods section). Synthetic waveforms show that such precursors are observable for a single event. Contrary to terrestrial studies[28–30], where stacks of hundreds of seismic records are needed to enhance the signal, stacking is not required owing to the relatively high velocity (or impedance) contrast across the intra-crustal interface on Mars (Supplementary Fig. 5).

The interface at the base of the second layer (at ~20 km) beneath the lander can generate PP and SS precursors with similar arrivals and amplitude as observed in the data (Fig. 3 and Supplementary Fig. 4). However, the signature of the shallowest layer seen at the landing site

(which should produce another strong precursor at around −6 s for the PP wave and −10 s for the SS wave), is absent from the recorded waveforms after polarization filtering. We also found that the velocity contrast across the base of the third layer (i.e., the Moho) is too small to generate significant precursors.

## Inverted crustal models beneath the bounce point

We performed inversions to constrain the crustal structure beneath the bounce point (see Methods section). We first conducted an inversion assuming only one crustal layer given that only one precursor with a relatively large amplitude is observed in both the PP and SS wave trains in the polarization-filtered data (Fig. 2). Figure 4c shows all the acceptable S-wave models (which fit both the travel time and amplitude measurements within one standard deviation) from this inversion. The depth of the inverted crustal interface is $23.3 \pm 4.9$ km (Fig. 4e), which is similar to that of the second intra-crustal interface at the landing site[11] (i.e., $20 \pm 5$ km). The derived S-wave speeds ($2.2 \pm 0.4$ km s$^{-1}$, in Fig. 4f) are also comparable to those from the receiver-function-derived values[11] ($2.3 \pm 0.3$ km s$^{-1}$).

The inverted models from the PP precursor exhibit a similar interface location of $17.6 \pm 2.5$ km (Fig. 4d, e). The derived P-wave speeds ($3.8 \pm 1.0$ km s$^{-1}$) also overlap with those beneath the lander in the second intra-crustal layer[11] ($4.1 \pm 0.5$ km s$^{-1}$, in Fig. 4g).

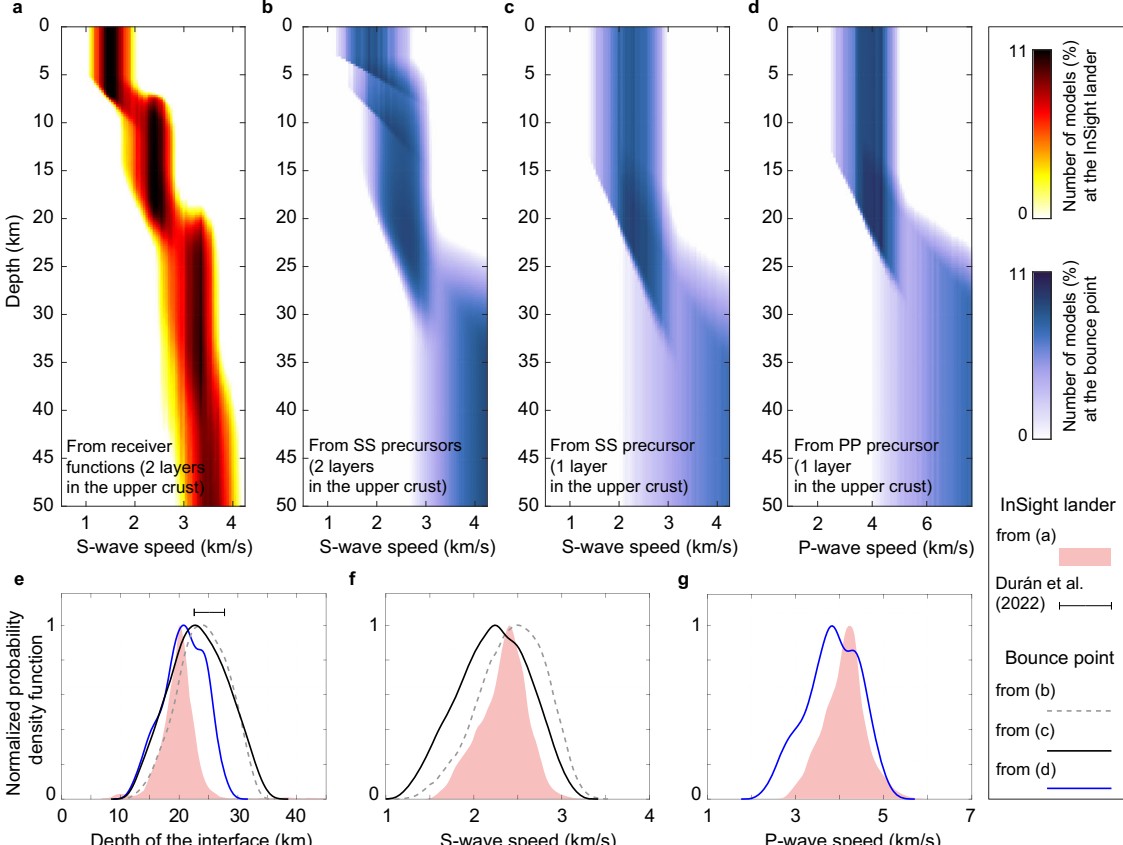

**Fig. 4 | Seismic velocity profiles from the inversion. a** The three-layer crustal models beneath the InSight landing site from the receiver function study[11], with an intra-crustal interface at 20 ± 5 km. The color scale indicates the number of models. **b** Inverted S-wave speed models at the bounce point with two layers in the upper crust from the SS precursors in this study. **c** Similar to (**b**), but with one layer in the upper crust. **d** Similar to (**c**), but for the P-wave speed with one layer in the upper crust. **e** Normalized probability density functions for the depth of the intra-crustal interfaces beneath the landing site (in shaded red) from the receiver function study[11], and at the bounce point from the precursor constraints (in solid and dashed curves). The inferred interface depth at the landing site from another study[13] is also shown as a horizontal bar. **f** Normalized probability density functions for the S-wave speed in the second layer beneath the lander (in shaded red), in the second layer for models in (**b**), and the first layer for models in (**c**) at the bounce point. **g** Similar to (**f**), but for the P-wave speed.

However, on the tangential component, we see other signals (e.g., at −5.0 s and −8.5 s) (Fig. 2a) that are mainly visible in the raw data but attenuated after the polarization filter is applied. We argue that such pulses are not likely to be the SS precursors because there are strong energies on the vertical component at the same arrival time (Supplementary Fig. 2a), which is not expected for shear waves (for the teleseismic event with nearly vertical incidence angle). Nevertheless, we performed another inversion assuming two crustal layers since if either of these two signals were to be another seismic precursor, another shallow interface beneath the bounce point would be required to generate this phase. Inversion results (Fig. 4b) show that adding an extra discontinuity in the top 12 km does not significantly change the location of the second interface (24.1 ± 4.2 km, in Fig. 4e) and the S-wave speeds within the layer (2.4 ± 0.4 km s⁻¹, in Fig. 4f).

We note that the velocity below the discontinuity is less-well constrained than the velocity above it. This is because the latter is constrained by both the differential arrival time and the amplitude ratio, whereas the former is only constrained by the amplitude ratio. In addition, the amplitude information is less reliable, partly due to the possible contamination from the local atmospheric energy injections[31] (Supplementary Fig. 7).

## Comparison between the landing site and the bounce point

The inversion results of this study (Fig. 4e, f), based either on the raw data or the polarization-filtered data, consistently show that the Martian crust at the bounce point shares a similar discontinuity at 18–24 km with the second intra-crustal interface beneath the InSight landing site, i.e., at ~20–25 km[11,13].

The shallowest interface (at 8 ± 2 km, discovered at the landing site), if present at the bounce point, should produce another strong precursor. Although energy is visible in the raw data between the main PP/SS phase and the major precursor (i.e., 19.3 s for SS and 10.2 s for PP), it is strongly attenuated after polarization filtering. This possibly indicates the absence of such a shallow layer at the bounce point, or that the velocity contrast across it is not as significant as beneath the lander. However, because of this inconsistency between the raw data and the polarization-filtered waveforms, we cannot confidently evaluate the regional variations in the uppermost crustal structure of Mars based on these results. Nevertheless, if the uppermost ~8 km layer is indeed absent or highly obscured at the bounce point location, it could result from a variety of factors, such as porosity differences, impact history, and/or younger volcanism, depending in part on its exact location (e.g., north or south of the dichotomy boundary; see below).

The third interface (i.e., the Moho) detected at the landing site has a limited impedance contrast. If this discontinuity exists at the precursor bounce point with similar depth and property as at the landing site, it cannot generate detectable SS or PP precursors (with an amplitude ratio of less than 20%, in Fig. 3 and Supplementary Fig. 4). Considering that the structure at the bounce point can be different, we also searched for the earlier precursors in a longer time window. However, we did not find any consistent precursors (i.e., the shape of the signal does not significantly change when back azimuth varies, and

the signal should show up in both the PP and SS wave trains) associated with the crust-mantle discontinuity (Supplementary Fig. 8).

## The "20 km" discontinuity on Mars

The estimation of the marsquake hypocenter location[7] results in a relatively small uncertainty on the corresponding position of the bounce point in the longitudinal direction, but a large uncertainty in the latitudinal direction (Fig. 1a). As a result, the location of the bounce point also covers a wide variety of geological features[32] (see also Supplementary Fig. 9), and thus leads to different geological interpretations of the seismic profiles. This region is transected by the crustal dichotomy boundary and thus includes heavily impacted Early- to Mid-Noachian crustal rocks to the south (that in places are covered (resurfaced) by Hesperian to Amazonian volcanic rocks), Hesperian to Amazonian transitional units at the boundary, and Hesperian to Late-Amazonian volcanic rocks to the north.

If the bounce point is situated to the north of the dichotomy boundary, the similar elevations (and the expected crustal structure) at the InSight lander and the possible bounce points[33] might explain the similar depth for the base of the crustal layer (i.e., ~20 km) at these two locations. Therefore, this 20-km interface could be a common discontinuity at least for regions near the dichotomy boundary. Alternatively, if the bounce point is located to the south of the dichotomy boundary, it might indicate that this 20-km interface is more likely a global feature since it has been observed in both the lowlands (i.e., the landing site) and the highlands (i.e., the bounce point).

For the origin of this "20 km" seismic interface at the bounce point, a variety of mostly non-mutually exclusive factors could play a role. Knapmeyer-Endrun et al. (2021) (ref. 11) and Wieczorek et al. (2022) (ref. 17) reviewed the possible crustal features and processes that might lead to a seismic velocity discontinuity at ~20 km depth at the landing site and they included the closure of porosity, presence of thick impact ejecta, long-lived magmatic activity, and large-scale compositional layering. However, the large uncertainties in absolute P- and S-wave speeds in the lower crust (i.e., beneath the discontinuity) coupled with uncertainty in the exact location of the bounce point limit the ability to interpret the exact cause or causes of the layering at the bounce point, although the similar scale and depth are consistent with a broadly common origin for the two locations.

Wieczorek et al. (2022) (ref. 17) favored a hypothesis involving the removal of pore space by viscous deformation at depth. Because porosity can significantly reduce both the P- and S-wave speeds[34,35] and the closure of pore space is expected to take place over only a few kilometers[36], the removal of pore space would generate a density (or seismic velocity) discontinuity. On the Moon, impact cratering has also been suggested as the cause for low seismic velocities or crustal porosity[37–39]. If it is the same on Mars, the last significant porosity-forming events should be related to the youngest basins (e.g., Argyre and Isidis, see Fig. 1a) which were formed during the Middle Noachian period[40]. Based on reasonable estimations of the surface heat flow at that time (i.e., about 3.9 Ga ago), Gyalay et al. (2020) (ref. 36) proposed that porosity should have been removed at depths greater than about 12–23 km since then. This range of the porosity-removal depth overlaps with the depth estimation of the "20 km" discontinuity beneath both the lander and the bounce point. If such an intra-crustal seismic interface reflects a transition from porous to non-porous Martian crustal materials, the velocity difference above and below the pore closure depth can be explained by a porosity reduction of about 10–16%, assuming a pore aspect ratio of 0.1 (see Methods section).

Although pore closure alone may explain a ~20 km discontinuity, an additional factor could be that the lower crust at the landing site and bounce point, on average, has a composition differing from the upper crust leading to increased P- and S-wave speeds. In general, an increase in the P- and S-wave speeds with depth would suggest a lower crust composed of more mafic to ultramafic lithologies compared to the overlying crust (e.g., ref. 41). Where younger volcanism was active after the formation of the ancient primary crust, significant amounts of highly mafic to ultramafic cumulate lithologies (up to ~50–70% of total magmatism, depending on water content) could reside in the lowermost crust[42,43] which could influence mean seismic velocities. Although the geology of the bounce point is uncertain (Supplementary Fig. 9), younger volcanism does play some role in the development of the crust at the landing site[44,45].

Since the first seismological constraints on the Martian crustal structure were obtained beneath the InSight lander[3,11], the nature of the crustal layers and whether they represent local geological structures or global features has been debated[17]. To answer this, we do not necessarily have to wait for additional seismometers to be placed on the Martian surface. Our analysis provides important constraints on this matter demonstrating that the "20 km" discontinuity, first discovered beneath the lander, is not a local geological structure but more likely a regional feature near the dichotomy boundary or possibly even a global feature.

To further investigate how the Martian crust varies from place to place, topography and gravity data (collected from orbit) were used to construct global Martian crustal thickness models by extrapolating the seismic measurements[17] (i.e., seismic "anchor points"). Thus far, only one such anchor point (located at the InSight landing site) is available, although the possible complexity of the first anchor point (e.g., one proposal is that the landing site stands above a plume head with a density anomaly[46]) might bias the estimations of global thickness models. A second anchor point at the precursor bounce point, with a more accurate location (either by the reduction of the location uncertainty or the utilization of the meteorite impact source) and a reasonably robust estimate of Moho depth, will effectively improve global crustal models of the red planet by removing most of the ambiguity (i.e., tradeoff) in the gravity and topography inversions.

## Methods
### Data processing
The data were originally recorded by the broadband SEIS seismometer on the U, V, and W components. We first pre-filtered the raw data from 0.01 to 8 Hz and removed the instrument response using ObsPy[47] to get the displacement records. We subsequently rotated the coordinates from UVW to NEZ and then from NE (i.e., North and East) to RT (i.e., Radial and Tangential directions) using the back azimuth determined by Horleston et al. (2022) (ref. 7). Finally, a two-pass, second-order Butterworth band-pass filter was applied to the waveforms on the Z, R, and T components (e.g., in Fig. 2, and Supplementary Fig. 1).

Since the SEIS seismometer is deployed under an extremely harsh environment, glitches (i.e., transient one-sided pulses) are generated due to internal thermal stresses resulting from the diurnal temperature changes and might interfere with the seismic signals[48]. Glitch detection (using a synthetic glitch template) found no glitches in the time windows around the PP and SS phases[7].

We also applied a time-domain polarization filter[24] to the data. This polarization filter has been successfully applied on Earth[27], Mars[49], and Moon[50] for enhancing teleseismic body wave signals. The inputs for this polarization filter are data on the Z, R, and T components, and it estimates the direction and rectilinearity of the particle motion. Then, the waveforms on all three components are weighted according to the rectilinearity since body waves exhibit a higher degree of linear polarization compared with noise. Finally, the waveform on each component is further weighted according to the direction of the particle motion[27]. We note that this weighting can explain why the signal seen from −5 s to −9 s on the tangential component of the raw data (Fig. 2a) is suppressed after polarization filtering (Fig. 2b). Specifically, at the relative arrival from −5 s to −9 s (relative to the SS arrival), the direction of the particle motion is dominated by the vertical component (e.g., the larger amplitude of the signal on the vertical component

than the one on the tangential component, as shown in Supplementary Fig. 2). Therefore, the last weighting will enhance the signal on the vertical component and weaken the signals on the tangential component at −8.5 s. We note that the down-weighting of this signal is justified because S-wave signals are less likely to occur on the vertical component.

## Comodulation analysis

Because InSight's seismometer is a surface-deployed sensor, it also measures the local atmospheric energy injected into the ground motion as noise. This noise injection requires an in-depth analysis that quantifies the contributions of the environmental signals to the seismic records[31]. The comodulation approach has been demonstrated to be particularly successful in identifying seismic energy above the expected broadband noise injected from the local weather[4,31,51]. Both pressure and wind-speed measurements by the on-deck mounted Auxiliary Payload Sensor Suite (APSS)[52] can be used to quantify this environmental noise injection. We also analyzed the excitation of the weather-sensitive lander mode at ~4 Hz as another effective atmospheric proxy in estimating this injection on Mars[31,53].

Comodulation analysis was applied to the seismic signal of S0976a to predict the power injected by atmospheric disturbances during the arrival of the PP and SS phases, and their respective precursors. Supplementary Fig. 7a shows that a clear excess seismic power, here taken as the sum of the signal variance in all three axes, is observed during the identified PP and SS arrivals between passbands centered at periods of 8 s and 1.4 s against the expected atmosphere-driven noise (measured both by the pressure sensor and weather-sensitive lander mode in all three directions). This excess energy is seen throughout the event before re-converging with the expected environmental noise injected into the system (measured by both the pressure power and weather-sensitive lander resonance at 4 Hz). For periods ≤1 s and ≥11.3 s, the broadband noise injected by the atmosphere dominates and the seismic signal becomes much weaker than pressure and lander-mode estimates, following closely the noise trajectory injected into the system over time. The main excursions observed in the passband centered at 11.3 s are glitches.

Supplementary Fig. 7b shows a similar comodulation analysis of the vertical component focused in a 200-s window centered at the PP arrival. The PP peak energy arrival appears in excess to the environment, most visibly in the 2–5.7 s period. The energy in the PP precursor is seen diverging from the match to the pressure or lander mode power, particularly in the frequency bands centered at 1/4 Hz and 1/5.7 Hz.

A comodulation analysis of the tangential component (rotated with a back azimuth of 101 degrees) is also shown in Supplementary Fig. 7c, focused in a 200 s window around the SS arrival. The SS energy arrival appears again to be in clear excess to the expected environmental injection below 2.8 s, with visible contamination in the higher frequency bands of 1/1.4 and 1/2 Hz. The energy in the SS precursor is seen diverging from the match to the pressure or lander mode power in the frequency bands centered from 1/2.8 Hz down to 1/8 Hz.

Based on the comodulation analysis, we chose 2.8 s to 6 s as the frequency band in this study (e.g., Fig. 2) where there is the least wind injection. We also considered relatively broader bands when measuring the arrival time and amplitude from the data (Supplementary Fig. 1).

## Forward modeling

We derived the differential travel time ($dt$) between the PP (or SS) phase and its precursor (if there is one layer, e.g., Fig. 4c) using ray theory:

$$dt = 2 * H * \sqrt{\frac{1}{V^2} - p^2}, \tag{1}$$

where $H$ is the thickness between the free surface and the crustal interface, $V$ is the average seismic wave speed in the crustal layer, and $p$ is the ray parameter.

When there are two crustal layers (e.g., Fig. 4b), the differential travel time ($dt$) is calculated using ray theory:

$$dt = 2 * H_1 * \sqrt{\frac{1}{V_1^2} - p^2} + 2 * (H_2 - H_1) * \sqrt{\frac{1}{V_2^2} - p^2}, \tag{2}$$

where $H_1$ and $H_2$, and $V_1$ and $V_2$ are the thicknesses and the corresponding seismic wave speeds for the first and second crustal layers, respectively.

The amplitude ratio was calculated according to the reflection and transmission coefficients[54,55] (also see Code Availability statement). We note that information on another parameter, i.e., density, was also needed for the amplitude calculation. We first calculated the density-to-Vs ratio using 20,000 models from the receiver function study[11], and then used its mean value of 0.8 (Supplementary Fig. 6c) to scale the density from the S-wave speed in each layer in this study. We also tested varying this ratio by one standard deviation (i.e., 0.05) in each layer and found the variation in density does not significantly change the inversion results (Supplementary Fig. 10).

The estimation of the ray parameter is dependent on both the location of the event and the velocity model. In this study, we chose 100 seismic velocity models from Stähler et al. (2021) (ref. 24). For each model, we also varied the focal depth from 0 km to 50 km and the epicentral distance from 146.3 − 6.9° to 146.3 + 6.9°, and then calculated the ray parameters using software Taup[56]. The average ray parameters for the S- and P-waves are 10.13 s deg⁻¹ and 4.84 s deg⁻¹, respectively (Supplementary Fig. 6). Finally, we chose three typical ray parameters to calculate the synthetic arrival time and amplitude ratio, i.e., the mean value (10.13 s deg⁻¹ for S-wave and 4.84 s deg⁻¹ for P-wave), the mean value plus one standard deviation (10.13 + 1.10 s deg⁻¹ for S-wave and 4.84 + 0.71 s deg⁻¹ for P-wave), and the mean value minus one standard deviation (10.13−1.10 s deg⁻¹ for S-wave and 4.84−0.71 s deg⁻¹ for P-wave).

For the synthetic waveform shown in Fig. 3, we extracted the source time function from the data after polarization filtering (i.e., from −3.2 to 5.5 s for PP and from −5.9 to 9.7 s for SS). To simulate the precursor generated at each interface, we first multiplied the amplitude of the source time function with the calculated synthetic amplitude ratio. Then, we shifted the source time function according to the differential travel time derived from Eqs. (1) and (2). We also compared this modeling approach with another software, QSEIS[57], which calculates the full wave field, and found a consistent amplitude ratio when there is no significant interference from other phases. If there are interferences (e.g., depth phases), since they show up after both the main SS phase and its precursor, they do not significantly affect the phase identification (Supplementary Fig. 11).

## Inversion

For the S-wave inversion (Fig. 4c), where we assumed one crustal layer, we performed grid searches for the depth of the interface (from 1 km to 55 km with an interval of 0.5 km), the S-wave speed in this layer (from 1.0 km s⁻¹ to 4.0 km s⁻¹ with an interval of 0.05 km s⁻¹), and the S-wave speed jump across the interface (from 0.1 km s⁻¹ to 3.0 km s⁻¹ with an interval of 0.05 km s⁻¹). Then, we calculated the synthetic travel time difference and amplitude ratio. We defined a model as being acceptable when both its synthetic differential arrival and amplitude ratio fall within one standard deviation of the measurement. The inversion scheme is similar for the P-wave case in Fig. 4d, with the P-wave speed in the layer changing from 1.8 km s⁻¹ to 7.2 km s⁻¹, and the P-wave speed jump to be from 0.2 km s⁻¹ to 5.4 km s⁻¹. The uncertainty of the inverted model parameters is defined as one standard deviation from all the acceptable models.

For the S-wave inversion (Fig. 4b), where we assumed two crustal layers, the thickness of the top layer varies from 1 km to 20 km, and the thickness of the second layer varies from 1 km to 35 km. The S-wave speed in the first (top) layer can vary from 1.0 km s$^{-1}$ to 3.0 km s$^{-1}$, and the S-wave speed jump across this intra-crustal interface and the second interface varies from 0.0 km s$^{-1}$ to 3.0 km s$^{-1}$ and from 0.1 km s$^{-1}$ to 3.0 km s$^{-1}$, respectively.

We note that for the inversions in Fig. 4, we limited the minimum values of the S- and P-wave velocities in the bottom layer (i.e., the lower half-space) to be no smaller than 2.0 km s$^{-1}$ for the S-wave and 3.6 km s$^{-1}$ for the P-wave (i.e., the maximum velocities in the shallowest crustal layer whose base is at ~8 km[11]). We also limited the maximum values of the S- and P-wave velocities in the bottom layer (i.e., the lower half-space) to be no larger than 4.3 km s$^{-1}$ for the S-wave and 7.7 km s$^{-1}$ for the P-wave (i.e., typical velocities in the uppermost mantle derived from seismic body wave inversions[24]). To test the dependence of the results on the priors, we updated those two maximum values to be 3.5 km s$^{-1}$ (for S-wave) and 6.2 km s$^{-1}$ (for P-wave), i.e., the mean velocities in the lower crust plus one standard deviation from all models in the receiver function study[11]. The updated values reflect the velocities in the lower crust, rather than the upper mantle. This is because it is reasonable to assume that this discontinuity at the bounce point is likely also an intra-crustal interface due to the similar depth of the "20 km" seismic interface compared with the landing site. Based on different priors, the new inversion results shown in Supplementary Fig. 12 are consistent with a discontinuity at ~20 km depth (i.e., 17.6 ± 2.5 km for PP-wave, 19.7 ± 3.5 km, and 20.2 ± 3.1 km for SS-wave).

The inversions presented in the main text try to fit the arrival time and amplitude ratio measurements within uncertainty (i.e., one standard deviation). For comparison, we also performed an inversion for the S-waves based on the misfit (i.e., the cross-correlation coefficient) between the polarization-filtered data and the synthetic waveforms (Supplementary Fig. 13). The inverted models based on the waveform fit do not significantly deviate from those based on the arrival time and amplitude ratio measurements, though the inverted depth of the interface is about 5 km deeper. This difference is partly because the left sidelobe of the main SS phase is not included in the travel time measurements (since this signal shows inconsistency between the raw and polarization-filtered waveforms) but is considered when calculating the waveform misfit. In this study, we prefer the inversion approach used in the main text where only reliable signals are picked, rather than the one based on the waveform misfit.

## Measurement and modeling uncertainties

We filtered both the raw data and polarization-filtered waveforms into different frequency bands (i.e., 1.5–6.0 s, 2.2–6.0 s, 2.8–6.0 s, 1.5–7.0 s, 2.2–7.0 s, 2.8–7.0 s) to estimate the uncertainties of both the differential arrival time and amplitude ratio (Supplementary Fig. 1). For the S-waves, we only picked the arrivals of the positive pulse and the right sidelobe of the main SS phase (i.e., triangles in Supplementary Fig. 1), since the left sidelobe shows inconsistency between the raw and polarization-filtered data. Then, we picked the corresponding pulses for the SS precursor (i.e., circles in Supplementary Fig. 1). We also picked the arrivals for the other two phases at −5 s (crosses in Supplementary Fig. 1) and −9 s (stars in Supplementary Fig. 1). The differential arrival and amplitude ratio are calculated at all the frequency bands for both the raw data and the polarization-filtered waveforms. Finally, we used the mean value as the measurement and one standard deviation as the uncertainty (Supplementary Fig. 3). For the P-waves, since the positive pulse for the PP precursor is contaminated by the wind noise (Supplementary Fig. 14), we picked the left negative sidelobe to measure the arrival time and amplitude. The final differential arrivals and amplitude ratios are 10.2 ± 0.2 s and 43.3 ± 7.5 % for PP, and 19.3 ± 0.5 s and 39.7 ± 16.1 % for SS.

Since the inverted depth range of the discontinuity at the bounce point overlaps with the second intra-crustal interface beneath the lander, the receiver-side reflection of the PP/SS precursor (though with a small amplitude) can interfere with the main PP/SS phase and reduce its amplitude by about 10% (Supplementary Fig. 11). Therefore, we added an extra uncertainty of ±10% to the amplitude ratio measurement to consider this possible interference.

## Locations of the Marsquake and the bounce point

Following standard procedures, as outlined in Clinton et al. (2021) (ref. 4), the marsquake service identified two distinct body wave phases for event S0976a. With a differential travel time of approximately 14 min, these phases cannot be direct P and S body phases and rather, were assigned as PP and SS[7]. The PP phase was identified in the time domain using a 2–6 s band-pass filter and was given an uncertainty of ±10 s. Similarly, the SS phase was also picked in the time domain but using a 2–8 s band-pass filter appropriate to the broader frequency range of this phase, and a ±10 s uncertainty was assigned to it too[7]. Note that this relatively large uncertainty of ±10 s was assigned in Horleston et al. (2022) (ref. 7) since there are some energies within this relatively long window (i.e., the PP/SS phases and the precursors identified in this study). Consistently, our preferred PP/SS picks are also within this uncertainty range (Fig. 2a).

Based on the PP and SS travel times (and their uncertainties)[7], the epicentral distance was calculated with 100 velocity models from Stähler et al. (2021) (ref. 24) using the probabilistic distance determination algorithm[58]. This gives this event the most likely epicentral distance as 146.3 ± 6.9º. The uncertainty in the distance arises from the uncertainty in the picks and the velocity models used.

The back azimuth was determined from the particle motion observed during the first few seconds of the PP arrival following Bose et al. (2017) (ref. 58). The back azimuth was determined from the amplitude ratio of the two horizontal components and was found to be 101 ± 25º. The amplitude ratio shows a clear peak at 101º, and the uncertainty is assigned by the azimuth at 25% of the peak[7].

The estimation of the marsquake hypocenter location leads to relatively a small uncertainty on the position of the bounce point in the longitudinal direction (from the uncertainty on the epicentral distance), but a large uncertainty in the latitudinal direction (from the uncertainty on the back azimuth). Because of finite frequency effects, the bounce point of the seismic wave is not a point (e.g., derived from ray theory) but a region of size related to the period of the seismic signal (e.g., ref. 59). To estimate the size of the Fresnel zone at the bounce point, we also computed the finite-frequency boundary kernel[60–62] for the PP precursor with a minimum resolvable period of ~5 s. The most sensitive part of the sensitivity kernel for a given marsquake location (i.e., the black star in Supplementary Fig. 15) is comparable to about 1/3 of the total size of the uncertainty of the bounce point location (i.e., the shaded white region, from the marsquake location error). We note that the size of the sensitivity kernel in Supplementary Fig. 15 is overestimated because of the relatively long period of 5 s we chose for the calculation due to the expensive computational cost. For the seismic data utilized in this study, the periods are shorter (i.e., the half durations of PP and SS phases are less than 2 s and 3 s, respectively), and thus, the size of the real sensitivity region will be smaller. Nevertheless, the more realistic uncertainty of the bounce point location, when considering the finite frequency effect, should be at most about 1/3 larger than the shaded white region in Supplementary Fig. 15. Given this relatively large uncertainty in the latitudinal direction, we cannot determine whether the "bounce point" is located in the lowlands, the highlands, or at the dichotomy boundary.

## Porosity effects

For the origin of the "20 km" seismic interface identified both beneath the landing site and the bounce point, a variety of mostly non-mutually

exclusive factors could play a role. An influential hypothesis is the removal of pore space at depth. To quantitively assess the influence of porosity, we assume that the velocity jump across the "20 km" interface is affected by porosity alone.

Above the "20 km" interface, the average P- and S-wave velocities beneath the InSight landing site are 4.1 km s$^{-1}$ and 2.3 km s$^{-1}$, respectively[11]. At the bounce point, the average P- and S-wave velocities above the interface are 3.8 km s$^{-1}$ and 2.4 km s$^{-1}$, respectively (i.e., Fig. 4). Below the interface, the average P- and S-wave velocities beneath the InSight landing site are 5.5 km s$^{-1}$ and 3.1 km s$^{-1}$, respectively[11]. At the bounce point, we prefer to use these lower mantle velocities inferred from the landing site, since the precursors are less able to resolve the structure beneath the discontinuity. This choice should be reasonable because Supplementary Fig. 12 indicates that there are no significant differences in the lower crust between those two places.

Then, we follow the scattering theory[63] to estimate both the P- and the S-wave velocity of porous basalt[64,65] as a function of porosity (Supplementary Fig. 16). In the calculation, a pore aspect ratio of 0.1 was assumed[34], and the pore space can be filled with gas (i.e., carbon dioxide) or liquid water (see Supplementary Table 1 for properties of the materials). We did not consider water ice as the pore-filling medium because temperatures are too high to freeze water at this depth range, near the equator[66,67].

If the pore spaces are removed below the "20 km" seismic interface, we should expect a corresponding porosity close to 0% in the lower crust. However, Supplementary Figs. 16a and 16c respectively show that, in the lower crust, an S-wave velocity of 3.1 km s$^{-1}$ corresponds to a porosity of ~10%, and a P-wave velocity of 5.5 km s$^{-1}$ corresponds to a porosity of ~8%. We note that this discrepancy is likely due to the assumption that the solid rock is 100% basalt in the calculation. Consistently, Wieczorek et al. (2022) (ref. 17) found that the density for known Martian igneous lithologies can vary by 25% relative and proposed that the Martian crust may be less mafic than many typical basaltic rocks, such as the shergottites, since the Martian crust, on average, has a grain density of <3100 kg m$^{-3}$. Therefore, we followed Kilburn et al. (2022) (ref. 67) and also considered plagioclase feldspar[68], the most common felsic mineral in basaltic systems. We would expect less mafic rocks in the Martian crust to have characteristics somewhere between these basalt-plagioclase end members (e.g., refs. 69, 70). Supplementary Fig. 16b, d show that, when considering plagioclase alone, the corresponding porosities in the lower crust (beneath the "20 km" discontinuity) are reduced to ~0% and ~5%, inferred from the S- and P-wave velocities, respectively.

Above the "20 km" discontinuity, Supplementary Fig. 16b, d also show that an S-wave velocity of 2.3–2.4 km s$^{-1}$ corresponds to a porosity of ~16%, and a P-wave velocity of 3.8–4.1 km s$^{-1}$ corresponds to a porosity of ~15% for the plagioclase end member. This indicates that the velocity difference above and below the pore closure depth could be explained by a porosity reduction of 10–16%, assuming a pore aspect ratio of 0.1.

## Data availability

The datasets generated during the analysis are available on the Zenodo repository: 10.5281/zenodo.7252035 and in the Supplementary Data 1 folder. The raw dataset used in this study is achieved and released by InSight Mars SEIS Data Service and is available to the science community (InSight Mars SEIS Data Service[71]) and is publicly available through the Planetary Data System (PDS) Geosciences node (InSight SEIS Data Bundle[72]), the Incorporated Research Institutions for Seismology (IRIS) Data Management Center under network code XB and the Datacenter of Institut de Physique du Globe, Paris.

## Code availability

All the computations made in this paper are either described in the method section or based on codes that are cited in the reference list.

The codes for data processing and structure inversion in this paper are available on the Zenodo repository: 10.5281/zenodo.7396882.

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

## Acknowledgements

We appreciate M. A. Wieczorek, A.-C. Plesa, and R. Maguire for their discussions. We acknowledge NASA, CNES, their partner agencies and Institutions (UKSA, SSO, DLR, JPL, IPGP-CNRS, ETHZ, IC, MPS-MPG), and the flight operations team at JPL, SISMOC, MSDS, IRIS-DMC, and PDS for providing SEED SEIS data. J.L. and C.B. are supported by NASA InSight PSP grant #80NSSC18K1679. S.M.M. is supported by NASA InSight PSP grant #80NSSC18K1622. P.L. is supported by Agence Nationale de la Recherche (ANR-19-CE31-0008-08 MAGIS, ANR-18-IDEX-0001 IdEx Université Paris Cité) and by CNES for SEIS science support. W.B.B. is supported by the InSight Project at the Jet Propulsion Laboratory, California Institute of Technology under a contract with the National Aeronautics and Space Administration (#80NM0018D0004). This is InSight Contribution Number ICN 246.

## Author contributions

J.L. designed the research and carried out the analysis, J.L., S.M.M., and C.B. discussed the results and proposed the interpretation, A.C.H. and G.Z. worked on the event location, C.C. and W.T.P. analyzed the wind injection, M.G. and V.L. validated the results, and Q.H. and E.B. calculated the sensitivity kernel. W.B.B. lead the InSight mission, P.L. is the PI of the SEIS instrument on InSight. J.L., C.B., and S.M.M. wrote the first draft, and all coauthors read and commented on the draft.

## Competing interests

The authors declare no competing interests.
