## [Peer Review File · Nature Communications]

Constraints on the Martian Crust Away From the InSight Landing SiteReviewer #1 (Remarks to the Author):

This study analyses the distant marsquake S0976a to identify both PP and SS precursors, which provides the first constraint on the Martian crustal structure away from the InSight landing site by using seismic observations. This allows for different geological interpretations to current models that are InSight-based. While this event was previously presented by Horleston et al., 2022, this study goes beyond by providing polarization analysis for time windows preceding the main PP- and SS-wave arrivals, and allows for the potential identification precursors. Although the main results of the paper are interesting, and the main points are clear, the paper is in need of considerable work and reorganization. For example, some of the modelling choices and assumptions made are not entirely clear until the final discussion is reached; most of the figures are unintelligible; and several points stated in the paper are devoid of explanation and references. In summary, I recommend major revisions. Below I list a number of points and questions that I would like the authors to consider when revising their manuscript.

- Lines 25 and 73: epicentral distance referred here is 146.3 +/- 6.9, however Horleston et al., (2022) claims 146 +/- 7. Please update with the correct value.

- Line 28: As stated later in the paper, due to the uncertainty on the epicentral distance of this marsquake, the location of the crustal structure "away", is not fixed, but model- and distance-dependent. Taking into account the uncertainty on the epicentral distance, a rough estimations tells me that the bouncing point is located between ~4100-4500km, and not fixed at 4000 km from the InSight landing site. Please clarify this in the abstract.

- Lines 31-33, and elsewhere: Please note that the study of Knapmeyer et al. (2021) was based on three events. There have been updates since that are based on 'many' more events (e.g., Duran et al. 2022). The consistent citation of a preliminary study seems a bit outdated as does the insistence on the two-layer crustal model. There is ample evidence that appears to favour a 3-layered crust. It would only be fitting if the authors were to briefly summarise the arguments/evidence in favour of this model and then maybe use that as 'anchoring point' for their study.

- Lines 34-35: "that has been obscured at the bounce point by later processes". This sentence is only understandable after reading the discussion. Please clarify in the abstract.

- Reference to InSight Marsquake Service 2022: catalog V10 cited as "upcoming". Meanwhile catalog V11 has been published. Please update.

- Lines 49-50: "and most of them are clustered at epicentral distances of about 30 degrees, i.e. 1700 km, to the east of the InSight landing site, near CF". This sentence is only true for the low-frequency family of marsquakes with known azimuth. High-frequency family of events have a different distribution of epicentral distances (e.g., see Giardini et al., 2020), and their azimuth is unknown.

- Lines 49-51: "and most of them are clustered at epicentral distances of about 30 degrees, i.e. 1700 km, to the east of the InSight landing site, near Cerberus Fossae (Fig 1c)". Please add reference for this or show the events in Fig. 1C.

- Fig. 1: Please increase all labels in the figure, they are really difficult to read. Particularly, all the annotations within the map are too small. It is generally so that any labels in a figure should have the same font size as the text in the figure caption. Please adhere to this 'rule'. Also, please highlight the different regions in the map that are mentioned in the text.

- Fig. 1C is referred to in the text before Figs. 1A and 1B. Also, Figs. 2A and 2B are referenced before Figs. 1A and 1B. Please rearrange text/figures so that a logical order

is followed.

- Fig. 1 caption: "This location is further narrowed down based on tectonic units (marked by the shaded region close to the Valles Marineris)". There is no reference or explanation of how the location is narrowed down based on tectonic units before the figure is presented. Please clarify this or consider the whole location in the analysis.

- Lines 74-75: "The epicenter is located near the western end of Valles Marineris (dashed area in Fig. 1c)". Related to my earlier comment about figure legibility, the dashed line in Fig. 1C is really difficult to distinguish.

- Line 77: "its radius is estimated at about 1855 km (Stähler et al., 2021)". Core radius in Stähler et al., (2021) is 1830 +/- 40 km, the value claimed here does not correspond to that study. Again the authors are referring to a preliminary estimate; other studies (e.g., Duran et al., 2022; Khan et al., 2022) have updated these values. Please update.

- Fig. 2: Please enhance figure quality. Labels and features are really difficult to read.

- Line 87: "The SS phase arrives 1847 s after the origin time...". This statement is model dependent. Which model was used to compute the origin time?

- Line 87: please add a reference to Fig. S8, where you analyzed the little influence the uncertainty on the back azimuth has on the computation of the tangential component.

- Lines 89-90: "all wave trains arriving before the SS phase are suppressed except for one signal at about -18.8 s relative to the SS phase". Not all the arrivals preceding the SS phase are suppressed, I also see arrivals between -40 and -30 s, and more at -10. In any case, their amplitudes are attenuated, but not suppressed.

- Line 126: "Our tests show that this layer should produce another strong precursor (Fig. 3c and d), but it is not observable in the polarization-filtered data (Fig. 2 and Fig. 3a)". I do observe a small precursor at ~10 s before the SS-wave train. The main results and main point of discussion of this paper are based on the "lack" of this phase, and, although small (which can be argued by a small velocity and density contrast on that crustal interface), this phase nevertheless seems to be present. Further explanation for why this phase is not considered is relegated to the final discussion (Lines 175-188), yet this should be included in the description of the observations and Figs. 2 and 3.

- Fig. 3: this figure shows synthetic waveforms of precursors using different crustal structure for comparison. Online methods offers an explanation about the computation of synthetic waveforms which does not require knowledge about the source mechanism. Although this method is a good first approximation, it does not take into account the arrival of additional waves not related to the precursors. Could you provide some more analysis considering waveform modeling to show if other arrivals are expected in the time window and that could lead to a misinterpretation of precursors?

- Fig. 3B: For the SS-wave train you compared the synthetics with the data after applying a polarization filter, while for the PP-wave you use raw data. Is there a reason not to use the polarized waveforms since they are expected to enhance linearly polarized waveforms? Please explain this in the text.

- Line 132: "only one clear precursor is observed in both the PP and SS wave trains". Again, this does not seem to be true if you consider the small amplitude arrivals preceding both waves.

- Line 133: "only one crustal layer is assumed in the inversion". Due to the observation of a single precursor in both wave trains, the inversion only accounts for one crustal layer. However, because of the other small-amplitude arrivals clearly visible in Figs. 3A and B, the inversion scheme should change and also consider more crustal layers as

done in previous studies (e.g., Knapmeyer-Endrun et al., 2021). A short comment about this is included in the final discussion, but results should also be shown.

- Line 134: "The best-fitting crustal thickness from the SS precursor is 22 ± 5 km". How are the uncertainties computed? Blue distributions in Fig. 4B seem wider than red distributions, but assigned uncertainties are the same. Please clarify.

- Line 134 and elsewhere: several times in the text you refer to crustal thickness, while in the figure caption this is crustal interface. I understand that the values ~ 20 km refer to crustal interface, and not to crustal thickness. Please correct.

- Lines 141-142: "We also found that the impedance contrast across the bottom interface of the third layer detected at the lander site is too small to generate any observable SS precursor". While this statement is true at the lander site, the structure at the bouncing point can be different, and indeed the goal of this paper is to constrain the crustal structure away from the InSight landing site. Therefore, a second test including both, differential arrival time and amplitude ratio, should also be considered to constrain the upper-most mantle velocity. Have you searched for earlier precursors that are potentially associated with the crust-mantle discontinuity?

- Fig. 4C: what does it mean "Acceptable models of the S-wave speeds"? Are these models the ones fitting the observations? Please explain the concept of "Acceptable". Also, the velocity structure shown in the plot is not clear, a plot where you show the individual models would depict the results better.

- Inversions: according to Online Methods, the amplitude ratios that are employed in the inversions are computed according to the reflection and transmission coefficients of Shearer (2009) and Aki and Richards (2002). These coefficients depend not only on the velocity, but also on the density. In the text, there is no mention of the density profile or any assumptions related hereto. Are the density values scaled to the velocity? Please clarify.

- Lines 166-167: "We note that, due to the relatively short period of the seismic signal (i.e., half-duration less than 3 s), the location of the bounce point estimated from ray theory does not deviate much from the more realistic case when considering finite frequency effects (Fig. S5 and Online Methods)". The referred "more realistic case" was computed using a single model and a single event location. For this case, the size of the bounce point results in a size comparable to the uncertainty of the location of the bounce point. However, both results are not comparable: one of them comes from the uncertainty on the source location, and the other from the fact that the bounce point is not a point but a region. Therefore, if you consider both cases, the final location of the bouncing point must be wider than shown in Fig. 1c. This statement should be reconsidered after computing the finite-frequency boundary kernels for several sources within the estimated source region.

- Line 211: "the absence of a shallow layer characterized by low seismic wave speed (and thus low density)". This sentence is not necessarily true. For both, P- and S-wave velocity, the relation with density is proportionally inverse (e.g., $V_s^2 = \mu/\rho$).

- The study provides two sets of independent inversions, one employing the SS precursors (Fig. 4C), and another employing PP precursors (Fig. 5C), which give slightly different results for the crustal interface depth: 22 ± 5 km and 20 ± 5 km, respectively. I would expect a joint inversion using both data sets to help constrain crustal interfaces better. Why is a joint inversion not undertaken.

- Horleston et al., (2022) refer to a second event that occurred on the far side of Mars with much higher SNR than S0976a. Why is this event not employed in the current study?

- Following previous comments, review all figures of Online methods to make them readable.

Reviewer #2 (Remarks to the Author):

Review of Nature Communications manuscript NCOMMS-22-26950 "Second Seismic Anchor Point of the Martian Crustal Structure away from the InSight Landing Site"

In their manuscript, the authors use data from the seismometer SEIS on NASA's InSight mission to Mars, specifically of the most distant marsquake as yet recorded, S0976a, to investigate the crustal structure at considerable distance from the InSight landing site. Since previous analyses only allowed constraining the crustal structure directly beneath the lander, this information is of special relevance for understanding variability in crustal structure and properties across Mars, and provides an additional, independent tie point for the global mapping of Mars' crust based on orbital measurements of gravity and topography.

In contrast to the seismic data of InSight used in previous publications, the large distance of S0976a allowed analyzing reflections of waves off discontinuities within the crust that arrive as precursors to the main phases reflected at the planet's surface. This is the first observation and interpretation of these phases on Mars, with significant implications, as outlined above. On the Moon, information on crustal thickness at multiple tie points could only be obtained by having multiple stations recording the same events, an approach that is not possible with just a single seismometer on Mars. Thus, other approaches are extremely valuable, even if – due to the sparsity of the martian seismic activity – they have to be based on the seismogram of a single event.

The authors find differences in crustal structure at the new location, about 4000 km distant from the lander, compared to what has been previously reported for the lander location. Specifically, they explain the data with just a single (intra-) crustal discontinuity around 20 km depth. This is the first seismic measurement that allows judging whether the crustal structure detected beneath the lander is truly globally representative, and thus very relevant. However, the analysis and interpretation steps of the manuscript at hand require some strengthening, improvement and removal of inconsistencies before publication. The authors can use only a single event, which, in the ray-theoretical framework they adopt, results in just one data point (travel time vs. relative amplitude) for the inversion, so special care has to be taken that to clarify all assumptions and use a consistent framework.

(1) The authors use a ray-theory based modelling, independently calculating the timing and relative amplitude of a possible precursor signal and then convolving this with the waveform of the direct PP or SS phase, as explained by equations (1) and (2) in the online methods. However, both travel times and amplitude ratios require information on the ray parameter/incidence angle, and the calculation of amplitudes also requires information on densities. The text does not mention how these were determined/selected. Thus, the information provided on the methods is insufficient to reproduce the results. Specifically, as Fig. 1 shows the error in event location, and there is also some uncertainty in the global Martian velocity models derived so far, especially for P-wave velocities in the lower mantle, how was the ray parameter chosen? At is the uncertainty and how does it affect the calculated travel times? And, in fact, what is the numerical value of the ray parameter? What assumptions are made on densities in the different layers? Are they kept constant in the inversion or is there a specific relation between densities and velocities? What is the reasoning behind this?

(2) Fig. 3 compares measured data to synthetics, but there are contradictions in the text describing what is actually shown and compared here. For SS, Fig. 3(a) shows the tangential component of the polarization filtered data. For PP, Fig. 3(b) shows the raw data of the vertical component without any additional filtering. Why do you choose to

show differently treated data in both cases? Are the synthetics in Figs. 3(c) - (f) also polarization filtered, or only some of them? I do not see why the two components should be treated differently here, specifically since l. 177ff says that conclusions are based on polarization-filtered data, and l. 360ff says the synthetics in Fig. 3 are based on polarization filtered data. Is this only true for SS or also for PP, or are raw PP data compared to polarization filtered synthetics (and why)?

(3) In Fig. 3, the authors show synthetics for a set of models derived from receiver functions in (c) and (d), and then the same synthetics when "the first layer in the model is excluded." What does this mean? Naively, to me that would imply just having one layer with the properties of layer 2. That's also similar to what the authors imply (l134ff.) But since the S-wave velocities in layer 2 are about 50% larger in layer 2, this would result in smaller travel-time for a constant total thickness of the layers according to eq. (2) and apparent in the higher velocities necessary for layer 2 in Extended Data Fig. 6 as compared to Fig. 4 to explain the same arrival. However, this is not the case in Fig. (3). Likewise, the amplitudes should be different – for one layer, there is just one pair of reflection/transmission coefficients to consider, whereas for two layers, the SS or PP wave is affected by two transmission coefficients, one for each layer, whereas the reflection coefficient between the bottom layer and the half-space below would stay the same. Again, this is not the case in Fig. (3). So what is actually done here? Keeping the same travel-time and relative amplitude for layer 2 and just not inserting the wave package for layer 1 in the calculation does not make much sense since removing layer 1 affects both relevant parameters for layer 2.

(4) In l. 141ff the authors state that the impedance contrast across the bottom interface in the previously derived models is too small to generate any observable SS precursor because the amplitude ratio is too small. However, they do not show any waveforms for this case that allow the reader to follow this conclusion. It would be good to also add synthetics for the three-layer case to Fig. (3), or at least show them – in comparison to the actual data – in the supplement.

(5) Along the same lines, some quantitative misfit would be useful in Fig. 3, i.e. the authors provide correlation coefficients in Extended Data Fig. 7 for the similarity between main phase and precursor. How about something similar for the differences between Fig. 3 (a) and (c) compared to (a) and (e), and the same for the PP waves?

(6) The authors give very clear arrival times for the SS (l. 87) and PP (l. 103) phases relative to the origin time, but state that MQS gave uncertainties of ± 10 s for both arrivals when locating the event (l. 404 and l. 406). Could you explicitly state the origin time you assume here – otherwise, it is hard for others to follow your phase identification. The ± 10 s uncertainty could potentially include both of your precursory phases. Could you show the MQS phase picks e.g. in Fig. 2? How do your precise values for the arrival times reflect back on the location, i.e. would this not reduce the location uncertainty shown in Fig. 1 significantly? You need to comment on this, including why you can reduce the timing uncertainty so much compared to MQS, in the text.

(7) The only discussion I found on the effect of a possible mantle triplication phase is in the caption of Extended Data Fig. 3. Nothing is said about this in the text. The figure caption does not specify how the triplication operator was calculated, what the corresponding velocity model looks like, what it is based on – i.e. some independent information from InSight or predictions from equations of state and mineralogy, or just the desire to reproduce the raw data shown at the bottom right of Fig. 3 more closely. What is the evidence for a mantle discontinuity "at around 1000 km" depth? This whole point needs some more elaboration.

(8) The inversion basically tries to fit two numbers within their uncertainty range, the differential arrival time of the phase and the relative amplitude. No waveforms are computed here, e.g. to visualize how well the computed results fit the actual data. This is simple, but possible, way to invert the data. In sharp contrast to this, one of the most

advanced modeling tools available is used to calculate the finite frequency effects for the size of the bounce point, SPECFEM3D_GLOBE, instead of e.g. looking at the Fresnel zone for SS and PP. How is this difference in approach warranted? What exactly are the benefits of using a 3D simulation here if the velocity model in the code is obviously 1D (i.e. it has a global Moho depth of 48 km, unlike the value proposed for the bounce point)? Why not run some full wavefield simulation at least for a preferred velocity model to see if it can reproduce the waveforms, if the tools are readily available?

(9) The authors state at several places in the text that the discontinuity at around 20 km depth they find is between the crust and mantle (i.e. l. 133, l. 138ff, where they say everything below 20 km is the mantle). Only in the abstract, they acknowledge that this could also be an intra-crustal discontinuity (l. 27). In the discussion, they point out the similarity between the discontinuity observed here and the discontinuity at about 20 km depth beneath the InSight lander (l. 170ff). However, the most recent publications point to this layer beneath the lander being within the crust, and not the crust-mantle interface (Duran et al. 2021, Kim et al. 2021, Wieczorek et al. 2022). If the Moho depth at the bounce point is in fact approximately 20 km, this would be another striking difference between the two locations, especially since gravity and topography predict similar crustal thickness assuming the same crustal density across the northern lowlands. This would mean that this assumption is fundamentally flawed, or the Moho depth at InSight is in fact 20 km. They authors need to discuss these implications, if they intend suggest a 20 km deep Moho – i.e. in how far does this modify the picture of the Martian crust derived in Wieczorek et al. (2022) in terms of origin and mineralogy of the crust, distribution of heat producing elements etc.?

(10) The geological discussion in the paragraph starting at l. 194 needs some rephrasing: the first sentence sounds like impact craters are related to the dichotomy, which is probably not what is meant. L. 200 – what's meant by "volcanic or sedimentary ash"? Doesn't ash always require a volcanic process to create it? L. 201 – please rephrase " a large impact cratering history" – is this a long history of impact cratering, a history of large impact craters, a cratering history with a large impact? L. 202 – Impact cratering has been suggested as the reason/cause for low seismic velocities on the Moon. L. 208 – what is meant with Noachian rocks that are resurfaced by Hesperian to Amazonian volcanic rocks? Hesperian and Amazonian volcanics deposited on top of Noachian rocks? Please clarify here and in l. 218.

(11) Finally, I am wondering about the title – if "Second seismic anchor point" is not somewhat hard to make sense of for the general reader. Something like "Martian crustal structure distant from the InSight landing site" or "Differences in Martian crustal structure at a location distant from the InSight landing site" or similar might be more accessible.

Minor points:

L. 34 needs to be clarified – like " indicating that this layer is either a local structure beneath the lander, or related to the type of geological unit, or it has been obscured at the bounce point by later processes."

It's also not clear what the difference between local structure beneath the lander and the type of geological unit is – in which other way could we define local structure on Mars than by looking at the geology?

L. 53 should be "using P-to-s receiver functions"

Fig. 1c: Line 74 states that the epicenter is given by the dashed area, but there are two dashed areas in Fig. 1c – please be more specific. The figure caption also talks about a fan-shaped dashed area that is narrowed down to a shaded region close to Valles Marineris – I can only discern one darkly shaded region around the black star indicating the epicenter and am at a loss to identify any fan-shape. Please try to mark and distinguish all the shaded areas on the map better.

L. 185ff – This sentence is misleading. The authors state a possible SS precursor observed at -8.5 s is more likely a P-wave since it is seen dominantly on the vertical component, which would not be expected for SS precursors. The next part of the

argument sounds like an incorrect back azimuth was somehow involved in transferring energy from the vertical (P-wave dominated) to the transverse (analyzed here for SS-precursors) component, which is not possible since the rotation for the backazimuth only involves radial and transverse components. I guess what's meant is spilling over of P-wave energy from the radial to the transverse component by either an incorrect backazimuth or anisotropy, but this has to be phrased more clearly – it is never indicated here that the radial component might also contain P-wave energy. Also, it would be helpful to show any figure of the radial component at all, e.g. include it in Extended Data Fig. 1, to get a full overview of the analyzed signal.

L. 325f: Reference to Fig. 1 is incorrect – there are no seismic waveforms shown in Fig. 1.

Reviewer #3 (Remarks to the Author):

This paper describes the identification of SS and PP precursors generated from the Martian crust-mantle discontinuity at a distance of 4000km from the InSight landing site using the most distant marsquake recorded by SEIS, S0976a. The authors further invert the observed data and obtain a crustal interface at a depth of 20 ± 5 km, in agreement with results from previous studies. However, a shallower interface at a depth of 8 ± 2 km previously reported at the InSight landing site by various studies was not observed at this location. The paper is well written, clear in its presentation, and certainly adds to our understanding of the Martian crustal structure. The techniques utilised are simple and elegant and primarily utilise polarization filtering of three component ground motion recordings along with forward and inverse modelling (i.e., grid-search approach). The authors then present their interpretation and reasoning for the discrepancy between the resulting structure at the bounce point and the structure at the landing site. With that said, the paper could be improved after taking into account the following concerns.

1) The authors show the raw and polarization filtered data for both the tangential and the vertical displacements in Figure 2. While the SS precursor phase is more evident in the tangential component, the PP precursor seen in the filtered vertical component appears less evident due to its seemingly lower amplitude ratio and possible phase distortion. It would be helpful to report this amplitude ratio for the PP precursor and the PP phase (and those derived from receiver function models) in the main text, as has been done for the SS precursor.

2) The possible second positive pulse in the PP precursor has been attributed to mantle triplication. This is presented quite clearly in Figure S3. However, it is not clear how the authors derive the mantle triplication operator without any knowledge of the deeper structure. The nature of the seismic discontinuity at 1000 km depth referred to here seems rather vague. Is it just an assumption, a possible inference, or has this been established by previous studies? Some more clarity about this is needed.

3) Figure 3 (a) shows polarisation filtered data for the tangential component, while Figure 3 (b) shows the raw displacement data for the vertical component. It would be preferable to use the polarization filtered data in both cases. There also seems to be no scale on the y axis in all the subplots. It would be a good idea to add it to have amplitude comparisons available.

4) In the Discussion section, the authors note that the polarization filter can suppress low amplitude signals that are more strongly affected by environmental noise. The authors are also aware of the contaminations observed in the seismic data recorded by

SEIS and confirm that there are no glitches around the observed PP and SS phases in So976a. However, the SEIS data also commonly shows strong environmental injections, as confirmed using data from the APSS (e.g., Charalambous et al., 2021). Have such correlations been taken into consideration by this study? The observations, if contaminated, can mimic phases that otherwise might not exist or show wrong polarization/amplitude properties that could lead to wrong conclusions (e.g., about the shallower discontinuity).

5) Although the analysis is impressive, these results have been based on just a single, small magnitude event. Terrestrial data has often shown that clear detection and identification of PP and SS precursors can be quite challenging even when a large number of events are available (e.g., Lessing et al., 2015). Stacking methods are therefore often employed. How do the authors explain this? A small discussion about this would certainly be helpful.

References

Charalambous, Constantinos, et al. "A comodulation analysis of atmospheric energy injection into the ground motion at InSight, Mars." *Journal of Geophysical Research: Planets* 126.4 (2021): e2020JE006538.

Lessing, Stephan, et al. "On the difficulties of detecting PP precursors." *Geophysical Journal International* 201.3 (2015): 1666-1681.

Thank you for your time and expertise with our manuscript. We appreciate the constructive feedback from the three reviewers. Below, we outline the reviewers' comments (in blue), and respond to each point-by-point (in black). New changes or additions to the manuscript are shown in red.

REVIEWER COMMENTS

Reviewer #1 (Remarks to the Author):

This study analyses the distant marsquake S0976a to identify both PP and SS precursors, which provides the first constraint on the Martian crustal structure away from the InSight landing site by using seismic observations. This allows for different geological interpretations to current models that are InSight-based. While this event was previously presented by Horleston et al., 2022, this study goes beyond by providing polarization analysis for time windows preceding the main PP- and SS-wave arrivals, and allows for the potential identification precursors. Although the main results of the paper are interesting, and the main points are clear, the paper is in need of considerable work and reorganization. For example, some of the modelling choices and assumptions made are not entirely clear until the final discussion is reached; most of the figures are unintelligible; and several points stated in the paper are devoid of explanation and references. In summary, I recommend major revisions. Below I list a number of points and questions that I would like the authors to consider when revising their manuscript.

A1: Thank you for the constructive suggestions. This time, we clearly described the modeling choices and assumptions, increased the quality of the figures, and added recent references.

- Lines 25 and 73: epicentral distance referred here is 146.3 ± 6.9 , however Horleston et al., (2022) claims 146 ± 7 . Please update with the correct value.

A2: We double-checked this value and verified that the epicentral distance in Table 1 of Horleston et al., (2022) is indeed 146.3 ± 6.9 . However, in the abstract of Horleston et al. (2022), they rounded 6.9 to 7. We prefer to keep the uncertainty of 6.9 since it has the same precision of decimal points as the mean epicentral distance of 146.3.

- Line 28: As stated later in the paper, due to the uncertainty on the epicentral distance of this marsquake, the location of the crustal structure “away”, is not fixed, but model- and distance-dependent. Taking into account the uncertainty on the epicentral distance, a rough estimations tells me that the bouncing point is located between ~4100-4500km, and not fixed at 4000 km from the InSight landing site. Please clarify this in the abstract.

A3: Thank you for pointing it out. We re-calculated the distance of the bounce point considering the uncertainty of the epicentral distance (146.3 ± 6.9). The distance should be between 4123 km and 4523 km, assuming a spherical Mars model without lateral heterogeneity. Therefore, we changed the previous statement from ‘4000 km’ to ‘about 4100 – 4500 km’.

We clarified this in the abstract:

“... which directly constrain the crustal structure away (**about 4100 - 4500 km**) from the InSight landing site.”

- Lines 31-33, and elsewhere: Please note that the study of Knapmeyer et al. (2021) was based on three events. There have been updates since that are based on 'many' more events (e.g., Duran et al. 2022). The consistent citation of a preliminary study seems a bit outdated as does the insistence on the two-layer crustal model. There is ample evidence that appears to favour a 3-layered crust. It would only be fitting if the authors were to briefly summarise the arguments/evidence in favour of this model and then maybe use that as 'anchoring point' for their study.

A4: In the previous manuscript (paragraph two), we cited more recent studies from Kim et al. (2021) and Durán et al. (2022) and mentioned that their studies favor the existence of the third layer. This time, we mentioned these two studies in the abstract, together with Knapmeyer et al. (2021).

We agree with the reviewer that the three-layer crustal model (i.e., a thicker crust) was supported by ample seismological studies (e.g., Kim et al. (2021) and Durán et al. (2022)), and we did not prefer to insist on the two-layer model.

This time, we particularly named the discontinuity at ~ 23 km as ‘intra-crustal interface’ throughout the manuscript and Fig. 1b to clarify this.

- Lines 34-35: “that has been obscured at the bounce point by later processes”. This sentence is only understandable after reading the discussion. Please clarify in the abstract.

A5: There were two conclusions in the previous version: conclusion 1 (the structures at the bounce point share a similar interface at ~ 20 km as the intra-crustal interface beneath the lander) is robust for both the raw data (Fig. 2a) and the polarization-filtered data (Fig. 2b); conclusion 2 (the structures at the bounce point do not likely show a shallow interface at ~ 8 km) is based on the polarization-filtered data. This time, we emphasize conclusion 1 and reduce the discussion of conclusion 2, and thus we removed this sentence in the abstract.

- Reference to InSight Marsquake Service 2022: catalog V10 cited as “upcoming”. Meanwhile catalog V11 has been published. Please update.

A6: We have updated the catalog version to V12.

- Lines 49-50: “and most of them are clustered at epicentral distances of about 30 degrees, i.e. 1700 km, to the east of the InSight landing site, near CF”. This sentence is only true for the low-frequency family of marsquakes with known azimuth. High-frequency family of events have a different distribution of epicentral distances (e.g., see Giardini et al., 2020), and their azimuth is unknown.

A7: Thank you for the correction. We have clarified this:

“In the first two and a half years, all the marsquakes with known epicentral distance occurred within a distance of fewer than 100 degrees (Horleston et al., 2022), and most of those with known back azimuth, are clustered at epicentral distances of about 30 degrees (i.e., about 1,700 km) to the east of the InSight landing site (Drilleau et al. 2021; Zenhäusern et al., 2022), near Cerberus Fossae (Fig 1a).”

- Lines 49-51: “and most of them are clustered at epicentral distances of about 30 degrees, i.e. 1700 km, to the east of the InSight landing site, near Cerberus Fossae (Fig 1c).”. Please add reference for this or show the events in Fig. 1C.

A8: We have added two references: Drilleau et al. (2021) and Zenhäusern et al., (2022). Please refer to #A7.

- Fig. 1: Please increase all labels in the figure, they are really difficult to read. Particularly, all the annotations within the map are too small. It is generally so that any labels in a figure should have the same font size as the text in the figure caption. Please adhere to this ‘rule’. Also, please highlight the different regions in the map that are mentioned in the text.

A9: Thank you for pointing it out. We have increased the labels size in all the figures and highlighted the regions in Fig. 1 that are mentioned in the text.

- Fig. 1C is referred to in the text before Figs. 1A and 1B. Also, Figs. 2A and 2B are referenced before Figs. 1A and 1B. Please rearrange text/figures so that a logical order is followed.

A10: We have rearranged the order of figures in Fig. 1.

- Fig. 1 caption: “This location is further narrowed down based on tectonic units (marked by the shaded region close to the Valles Marineris)”. There is no reference or explanation of how the location is narrowed down based on tectonic units before the figure is presented. Please clarify this or consider the whole location in the analysis.

A11: Yin (2022) proposed strike-slip faults in the Valles Marineris. However, there is no direct link between this event (without focal depth information) and the faults. Therefore, we agree with the reviewer that it is better to remove this argument and use the whole location uncertainty from the seismological constraints.

- Lines 74-75: “The epicenter is located near the western end of Valles Marineris (dashed area in Fig. 1c)”. Related to my earlier comment about figure legibility, the dashed line in Fig. 1C is really difficult to distinguish.

A12: We have greatly increased the figure legibility, and attached the high-resolution figures.

- Line 77: “its radius is estimated at about 1855 km (Stähler et al., 2021)”. Core radius in Stähler et al., (2021) is 1830 +/- 40 km, the value claimed here does not correspond to that study. Again the authors are referring to a preliminary estimate; other studies (e.g., Duran et al., 2022; Khan et al., 2022) have updated

these values. Please update.

A13: We added references to these studies, e.g., 1790-1870 km from Stähler et al., (2021), 1820-1870 km from Duran et al., (2022), and 1830-1850 km from Khan et al., (2022), and a new reference Wang and Tkalčić (2020) with a radius of 1792-1832 km. We also updated the text using the union of the value from these three works:

“with a radius estimated at about 1790-1870 km (e.g., Stähler et al., 2021; Duran et al., 2022; Khan et al., 2022; Wang and Tkalčić, 2022).”

- Fig. 2: Please enhance figure quality. Labels and features are really difficult to read.

A14: We have enhanced increased the figure quality and attached the high-resolution figures.

- Line 87: “The SS phase arrives 1847 s after the origin time...”. This statement is model dependent. Which model was used to compute the origin time?

A15: This time, we specified the origin time to be 2021-08-25 03:32:20, from Horleston et al. (2022). We also added more content in the Online Method section to describe the representative models used in Horleston et al. (2022).

- Line 87: please add a reference to Fig. S8, where you analyzed the little influence the uncertainty on the back azimuth has on the computation of the tangential component.

A16: We added the reference to Fig. S8 (now it is Fig. S1).

- Lines 89-90: “all wave trains arriving before the SS phase are suppressed except for one signal at about -18.8 s relative to the SS phase”. Not all the arrivals preceding the SS phase are suppressed, I also see arrivals between -40 and -30 s, and more at -10. In any case, their amplitudes are attenuated, but not suppressed.

A17: We agree with the reviewer and replaced ‘suppressed’ with ‘attenuated’.

- Line 126: “Our tests show that this layer should produce another strong precursor (Fig. 3c and d), but it is not observable in the polarization-filtered data (Fig. 2 and Fig. 3a)”. I do observe a small precursor at ~10 s before the SS-wave train. The main results and main point of discussion of this paper are based on the “lack” of this phase, and, although small (which can be argued by a small velocity and density contrast on that crustal interface), this phase nevertheless seems to be present. Further explanation for why this phase is not considered is relegated to the final discussion (Lines 175-188), yet this should be included in the description of the observations and Figs. 2 and 3.

A18: We agree with the reviewer that this phase, although attenuated, is present. Therefore, this time, we

emphasized more on conclusion 1, and reduced the discussions on conclusion 2 (please refer to #A5).

In addition, we included another inversion with two crustal layers, and found that adding an extra shallow layer does not much change the location of the second discontinuity at ~ 20 km. We added this inversion results in Fig. 4.

“However, on the tangential component, there are other signals (e.g., at -5.0 s and -8.5 s) (Fig. 2a) that are mainly visible in the raw data but much attenuated after the polarization filtering. We propose such pulses are not likely to be the SS precursors because there are strong energies on the vertical component at the same arrival time (Fig. S2a), which should not be expected for the shear waves.

Nevertheless, we performed another inversion assuming two crustal layers since if either of these two signals were to be another seismic precursor, another shallow interface beneath the bounce point would be required to generate this phase. Inversion results (Fig. 4b) show that adding an extra discontinuity in the top 12 km does not significantly change the location of the second layer (24.0 ± 4.4 km, in Fig. 4e) and the S-wave speeds within it (2.4 ± 0.4 km/s, in Fig. 4f).”

- Fig. 3: this figure shows synthetic waveforms of precursors using different crustal structure for comparison. Online methods offers an explanation about the computation of synthetic waveforms which does not require knowledge about the source mechanism. Although this method is a good first approximation, it does not take into account the arrival of additional waves not related to the precursors. Could you provide some more analysis considering waveform modeling to show if other arrivals are expected in the time window and that could lead to a misinterpretation of precursors?

A19: We added a comparison between this modeling approach and another software, QSEIS which calculates the full wave field. The related discussions can be found in the Online Methods (see also Fig. S11). The main conclusion is that the amplitude ratios between those two approaches are consistent when there is no significant interference from other phases. Even if there is interference (e.g., depth phases), since it arrives after both the main SS phase and its precursor, it does not significantly affect the phase identification.

- Fig. 3B: For the SS-wave train you compared the synthetics with the data after applying a polarization filter, while for the PP-wave you use raw data. Is there a reason not to use the polarized waveforms since they are expected to enhance linearly polarized waveforms? Please explain this in the text.

A20: This time, to be more consistent, we show the polarization filtered data (in red) for both PP and SS waves in Fig. 3. In addition, we plotted the raw data in light grey color for comparison.

- Line 132: “only one clear precursor is observed in both the PP and SS wave trains”. Again, this does not seem to be true if you consider the small amplitude arrivals preceding both waves.

A21: We corrected the statement:

“Given the fact that only **one precursor with relatively large amplitude** is observed in both the PP and SS wave trains in the polarization-filtered data (Fig. 3)”

- Line 133: “only one crustal layer is assumed in the inversion”. Due to the observation of a single precursor in both wave trains, the inversion only accounts for one crustal layer. However, because of the other small-amplitude arrivals clearly visible in Figs. 3A and B, the inversion scheme should change and also consider more crustal layers as done in previous studies (e.g., Knapmeyer-Endrun et al., 2021). A short comment about this is included in the final discussion, but results should also be shown.

A22: We added the results in Fig. 4 and added the content of the related discussion (please refer to #A18).

- Line 134: “The best-fitting crustal thickness from the SS precursor is 22 ± 5 km”. How are the uncertainties computed? Blue distributions in Fig. 4B seem wider than red distributions, but assigned uncertainties are the same. Please clarify.

A23: This time, we updated the figures (Fig. 4 and 5) and values, and also clarified this in the manuscript:

“The uncertainty of the inverted model parameters is defined as one standard deviation from all the acceptable models.”

- Line 134 and elsewhere: several times in the text you refer to crustal thickness, while in the figure caption this is crustal interface. I understand that the values ~20 km refer to crustal interface, and not to crustal thickness. Please correct.

A24: We have corrected it.

- Lines 141-142: “We also found that the impedance contrast across the bottom interface of the third layer detected at the lander site is too small to generate any observable SS precursor”. While this statement is true at the lander site, the structure at the bouncing point can be different, and indeed the goal of this paper is to constrain the crustal structure away from the InSight landing site. Therefore, a second test including both, differential arrival time and amplitude ratio, should also be considered to constrain the upper-most mantle velocity. Have you searched for earlier precursors that are potentially associated with the crust-mantle discontinuity?

A25: We added an additional search for earlier precursors (Fig. S7). However, we found no consistent precursor (i.e., the shape of the signal changes when back azimuth varies, or signal does not show up in both the PP and SS wave trains).

“The third layer (i.e., the Moho) detected at the lander site has limited impedance contrast. If it exists at the bounce point with similar depth and property, it could not generate detectable SS or PP precursors (with an amplitude ratio of less than 20%, in Fig. 3 and S4). Considering the structure at the bounce point can be different, we also searched for earlier precursors in a longer time window. However, we did not find any

consistent precursors (i.e., the shape of the signal does not significantly change when back azimuth varies, and the signal should show up in both the PP and SS wave trains) associated with the crust-mantle discontinuity (Fig. S7).”

- Fig. 4C: what does it mean “Acceptable models of the S-wave speeds”? Are these models the ones fitting the observations? Please explain the concept of “Acceptable”. Also, the velocity structure shown in the plot is not clear, a plot where you show the individual models would depict the results better.

A26: We have explained it in the manuscript:

“We defined a model as being acceptable when both its synthetic differential arrival and amplitude ratio fall within the measurement plus or minus one standard deviation.”

We tried to plot the individual models but found that plot is too busy. Therefore, this time, we changed the color scale and added normalized probability density functions to better illustrate them (Fig. 4).

- Inversions: according to Online Methods, the amplitude ratios that are employed in the inversions are computed according to the reflection and transmission coefficients of Shearer (2009) and Aki and Richards (2002). These coefficients depend not only on the velocity, but also on the density. In the text, there is no mention of the density profile or any assumptions related hereto. Are the density values scaled to the velocity? Please clarify.

A27: We added an additional test on the effect of density (Fig. S10), and related discussion in both the Online Methods part.

“The amplitude ratio was calculated according to the reflection and transmission coefficients from Shearer (2009) and Aki and Richards (2002). We note that information on another parameter, i.e., density, was also needed for the amplitude calculation. We first calculated the density-to- V_s ratio from 10,000 models in the receiver function study (Knapmeyer-Endrun et al., 2021), and then used its mean value of 0.8 (Fig. S6c) to scale the density from the S-wave speed in each layer in this study. We also tested varying this ratio by one standard deviation (i.e., 0.05) in each layer and found the variation in density does not significantly change the inversion results (Fig. S10).”

- Lines 166-167: “We note that, due to the relatively short period of the seismic signal (i.e., half-duration less than 3 s), the location of the bounce point estimated from ray theory does not deviate much from the more realistic case when considering finite frequency effects (Fig. S5 and Online Methods)”. The referred “more realistic case” was computed using a single model and a single event location. For this case, the size of the bounce point results in a size comparable to the uncertainty of the location of the bounce point. However, both results are not comparable: one of them comes from the uncertainty on the source location, and the other from the fact that the bounce point is not a point but a region. Therefore, if you consider both cases, the final location of the bouncing point must be wider than shown in Fig. 1c. This statement should be reconsidered after computing the finite-frequency boundary kernels for several sources within the estimated source region.

A28: We agree with the reviewer that those two uncertainties (one from the location error, and the other from the finite frequency effect) should be accounted for. We have clarified it in the Online Methods:

“The most sensitive part of the sensitivity kernel for a given marsquake location (i.e., the black star in Fig. S14) is comparable to about 1/3 of the total size of the uncertainty of the bounce point location (i.e., the shaded white region, from the marsquake location error). We note that the size of the sensitivity kernel in Fig. S14 is overestimated because of the relatively long period of 5 s we chose for the calculation due to the expensive computational cost. For the seismic data utilized in this study, the periods are shorter (i.e., the half durations of PP and SS phases are less than 2 s and 3 s, respectively), and thus, the size of the real sensitivity region will be smaller. Nevertheless, the more realistic uncertainty of the bounce point location, when considering the finite frequency effect, should be, at most, about 1/3 larger than the shaded white region in Fig. S14. Given this relatively large uncertainty along the latitude direction, the bounce point could be in the lowlands, the highlands, or at the dichotomy boundary.”

- Line 211: “the absence of a shallow layer characterized by low seismic wave speed (and thus low density)”. This sentence is not necessarily true. For both, P- and S-wave velocity, the relation with density is proportionally inverse (e.g., $V_s^2 = \mu/\rho$).

A29: We agree with the reviewer that density is inversely proportional to V_s^2 . But, the shear modulus also goes up as a function of density (and V_p). The empirical evidence is that for a broad range of compositions (including basaltic compositions), density increases as a function of V_s . In the current version, we removed this sentence since it is related to conclusion 2 (we emphasized on conclusion 1, please refer to #A5).

- The study provides two sets of independent inversions, one employing the SS precursors (Fig. 4C), and another employing PP precursors (Fig. 5C), which give slightly different results for the crustal interface depth: 22 +/- 5 km and 20 +/- 5 km, respectively. I would expect a joint inversion using both data sets to help constrain crustal interfaces better. Why is a joint inversion not undertaken.

A30: We agree with the reviewer that it is a great idea to perform a joint inversion. However, based on the comodulation analysis, PP precursor is contaminated by the wind noise (Fig. S13) in certain frequency bands. Although we chose a frequency band that is the least affected by the wind injection, the PP results might be less reliable compared to the SS results. Therefore, we prefer not to perform a joint inversion. Nevertheless, we added new figures showing the normalized probability density function for both the PP and SS results (Fig. 4e and 4f). In such figures, the ‘intersection’ of the PP and SS results can be viewed as results from a joint inversion.

- Horleston et al., (2022) refer to a second event that occurred on the far side of Mars with much higher SNR than S0976a. Why is this event not employed in the current study?

A31: The main reason for our choice to only use S0976a is that event S1000a is a complex event. For example, though S1000a shows higher signal-to-noise ratio than S0976a (the one we used in this study), the uncertainty of the PP and SS arrivals for S1000a are ± 20 s and ± 60 s in Horleston et al., (2022), which

is much larger than those for S0976a (± 10 s). In addition, this event shows unprecedented energy at all frequencies (Horleston et al. 2022). Although S1000a has been confirmed as an impact, the body wave signals are more complex and difficult to disentangle. Therefore, we prefer not to include this other event at this point.

- Following previous comments, review all figures of Online methods to make them readable.

A32: We have greatly increased the figure legibility, and attached the high-resolution figures.

Reviewer #2 (Remarks to the Author):

Review of Nature Communications manuscript NCOMMS-22-26950 “Second Seismic Anchor Point of the Martian Crustal Structure away from the InSight Landing Site”

In their manuscript, the authors use data from the seismometer SEIS on NASA’s InSight mission to Mars, specifically of the most distant marsquake as yet recorded, S0976a, to investigate the crustal structure at considerable distance from the InSight landing site. Since previous analyses only allowed constraining the crustal structure directly beneath the lander, this information is of special relevance for understanding variability in crustal structure and properties across Mars, and provides an additional, independent tie point for the global mapping of Mars’ crust based on orbital measurements of gravity and topography.

In contrast to the seismic data of InSight used in previous publications, the large distance of S0976a allowed analyzing reflections of waves off discontinuities within the crust that arrive as precursors to the main phases reflected at the planet’s surface. This is the first observation and interpretation of these phases on Mars, with significant implications, as outlined above. On the Moon, information on crustal thickness at multiple tie points could only be obtained by having multiple stations recording the same events, an approach that is not possible with just a single seismometer on Mars. Thus, other approaches are extremely valuable, even if – due to the sparsity of the martian seismic activity – they have to be based on the seismogram of a single event.

The authors find differences in crustal structure at the new location, about 4000 km distant from the lander, compared to what has been previously reported for the lander location. Specifically, they explain the data with just a single (intra-) crustal discontinuity around 20 km depth. This is the first seismic measurement that allows judging whether the crustal structure detected beneath the lander is truly globally representative, and thus very relevant. However, the analysis and interpretation steps of the manuscript at hand require some strengthening, improvement and removal of inconsistencies before publication. The authors can use only a single event, which, in the ray-theoretical framework they adopt, results in just one data point (travel time vs. relative amplitude) for the inversion, so special care has to be taken that to clarify all assumptions and use a consistent framework.

(1) The authors use a ray-theory based modelling, independently calculating the timing and relative amplitude of a possible precursor signal and then convolving this with the waveform of the direct PP or SS

phase, as explained by equations (1) and (2) in the online methods. However, both travel times and amplitude ratios require information on the ray parameter/incidence angle, and the calculation of amplitudes also requires information on densities. The text does not mention how these were determined/selected. Thus, the information provided on the methods is insufficient to reproduce the results. Specifically, as Fig. 1 shows the error in event location, and there is also some uncertainty in the global Martian velocity models derived so far, especially for P-wave velocities in the lower mantle, how was the ray parameter chosen? At is the uncertainty and how does it affect the calculated travel times? And, in fact, what is the numerical value of the ray parameter? What assumptions are made on densities in the different layers? Are they kept constant in the inversion or is there a specific relation between densities and velocities? What is the reasoning behind this?

A33: Thank you for pointing this out. We added more details on how we chose the ray parameters (based on 100 mantle velocity models and the possible event locations) and added a test on the dependence on density. Please also refer to #A27.

“The estimation of the ray parameter is dependent on both the location of the event and the velocity model. In this study, we chose 100 seismic velocity models from Stähler et al. (2021). For each model, we also varied the focal depth from 0 km to 50 km and the epicentral distance from $146.3 - 6.9$ to $146.3 + 6.9$, and then calculated the ray parameter using software Taup (Crotwell et al., 1999). The average ray parameters for the S- and P-waves are 10.13 s/deg and 4.84 s/deg, respectively (Fig. S6). Finally, we chose three typical ray parameters to calculate the synthetic arrival time and amplitude ratio, i.e., the mean value (10.13 s/deg for S-wave and 4.84 s/deg for P-wave), the mean value plus one standard deviation ($10.13 + 1.10$ s/deg for S-wave and $4.84 + 0.71$ s/deg for P-wave), and the mean value minus one standard deviation ($10.13 - 1.10$ s/deg for S-wave and $4.84 - 0.71$ s/deg for P-wave).”

“The amplitude ratio was calculated according to the reflection and transmission coefficients from Shearer (2009) and Aki and Richards (2002). We note that information on another parameter, i.e., density, was also needed for the amplitude calculation. We first calculated the density-to- V_s ratio from 10,000 models in the receiver function study (Knapmeyer-Endrun et al., 2021), and then used its mean value of 0.8 (Fig. S6c) to scale the density from the S-wave speed in each layer in this study. We also tested varying this ratio by one standard deviation (i.e., 0.05) in each layer and found the variation in density does not significantly change the inversion results (Fig. S10).”

(2) Fig. 3 compares measured data to synthetics, but there are contradictions in the text describing what is actually shown and compared here. For SS, Fig. 3(a) shows the tangential component of the polarization filtered data. For PP, Fig. 3(b) shows the raw data of the vertical component without any additional filtering. Why do you choose to show differently treated data in both cases? Are the synthetics in Figs. 3(c) - (f) also polarization filtered, or only some of them? I do not see why the two components should be treated differently here, specifically since l. 177ff says that conclusions are based on polarization-filtered data, and l. 360ff says the synthetics in Fig. 3 are based on polarization filtered data. Is this only true for SS or also for PP, or are raw PP data compared to polarization filtered synthetics (and why)?

A34: This time, to be more consistent, we show the polarization filtered data (in red) for both PP and SS waves in Fig. 3. In addition, we plotted the raw data in light grey color for comparison. We have also clarified the relationship between the data and the conclusions:

- Conclusion 1 (the structures at the bounce point share a similar interface at ~ 20 km as the intra-crustal interface beneath the lander) is robust for both the raw and the polarization-filtered data.
- Conclusion 2 (the structures at the bounce point do not likely show a shallow at ~ 8 km) is based on the polarization-filtered data.

We also added details on how the synthetic waveforms were generated (they are not polarization filtered):

“For the synthetic waveform shown in Fig. 3, we extracted the source time function from the data after the polarization filtering (i.e., from -5.9 to 9.7 s for SS and from -3.2 to 5.5 s for PP). To simulate the precursor generated at each interface, we first multiplied the amplitude of the source time function with the calculated synthetic amplitude ratio. Then, we shifted the source time function according to the differential travel time derived from equations (1) and (2).”

(3) In Fig. 3, the authors show synthetics for a set of models derived from receiver functions in (c) and (d), and then the same synthetics when “the first layer in the model is excluded.” What does this mean? Naively, to me that would imply just having one layer with the properties of layer 2. That’s also similar to what the authors imply (1134ff.) But since the S-wave velocities in layer 2 are about 50% larger in layer 2, this would result in smaller travel-time for a constant total thickness of the layers according to eq. (2) and apparent in the higher velocities necessary for layer 2 in Extended Data Fig. 6 as compared to Fig. 4 to explain the same arrival. However, this is not the case in Fig. (3). Likewise, the amplitudes should be different – for one layer, there is just one pair of reflection/transmission coefficients to consider, whereas for two layers, the SS or PP wave is affected by two transmission coefficients, one for each layer, whereas the reflection coefficient between the bottom layer and the half-space below would stay the same. Again, this is not the case in Fig. (3). So what is actually done here? Keeping the same travel-time and relative amplitude for layer 2 and just not inserting the wave package for layer 1 in the calculation does not make much sense since removing layer 1 affects both relevant parameters for layer 2.

A35: Thank you for pointing this out. We agree with the reviewer that the removal of the first layer in the receiver-function models requires more consideration, and this removal might not be necessarily needed. This time, we decided to only show the synthetics from the original receiver-function models, together with both the raw data and polarization-filtered data. Please refer to the updated Fig. 3.

(4) In l. 141ff the authors state that the impedance contrast across the bottom interface in the previously derived models is too small to generate any observable SS precursor because the amplitude ratio is too small. However, they do not show any waveforms for this case that allow the reader to follow this conclusion. It would be good to also add synthetics for the three-layer case to Fig. (3), or at least show them – in comparison to the actual data – in the supplement.

A36: Previously, we indeed added the synthetic waveforms from the bottom interface in Fig. 3. However, because of its smaller amplitude (and uncertainty in the models), those waveforms are too weak to be

observed. This time, we changed the color scale to make those waveforms clearer, and added annotations for the phases. Please refer to the updated Fig. 3.

(5) Along the same lines, some quantitative misfit would be useful in Fig. 3, i.e. the authors provide correlation coefficients in Extended Data Fig. 7 for the similarity between main phase and precursor. How about something similar for the differences between Fig. 3 (a) and (c) compared to (a) and (e), and the same for the PP waves?

A37: In the new version of the paper, we added auto-correlation coefficients in Fig. 2 to emphasize the similarity between the main phase and its precursor. We also performed an additional inversion based on the waveform misfit.

“The inversions, in the main text, try to fit the arrival time and amplitude ratio measurements within their uncertainty range (i.e., one standard deviation). We have also performed another inversion, for the S-waves, based on the misfit (i.e., the cross-correlation coefficient) between the polarization-filtered data and the synthetic waveforms for comparisons (Fig. S12). The inverted models based on the waveform misfit do not significantly deviate from those from the arrival time and amplitude ratio measurements, though the inverted depth of the interface is about 5-km deeper. This slightly greater depth of the interface from the waveform misfit inversion is understandable since the left sidelobe of the main SS phase is also included. However, this pulse is not considered in the travel time measurements since this signal shows inconsistency between the raw and polarization-filtered waveforms. Therefore, we prefer the inversion approach, in the main text, where only the reliable signals are picked.”

(6) The authors give very clear arrival times for the SS (l. 87) and PP (l. 103) phases relative to the origin time, but state that MQS gave uncertainties of ± 10 s for both arrivals when locating the event (l. 404 and l. 406). Could you explicitly state the origin time you assume here – otherwise, it is hard for others to follow your phase identification. The ± 10 s uncertainty could potentially include both of your precursory phases. Could you show the MQS phase picks e.g. in Fig. 2? How do your precise values for the arrival times reflect back on the location, i.e. would this not reduce the location uncertainty shown in Fig. 1 significantly? You need to comment on this, including why you can reduce the timing uncertainty so much compared to MQS, in the text.

A38: We added the MQS phase picks in Fig. 2. Our preferred PP/SS picks are within this uncertainty range. We also added a discussion on the event location uncertainty. The new pick will give a more precise epicentral distance but will not affect the back azimuth estimation much since the MQS phase pick includes our PP and PP precursor energy. Therefore, it will not significantly reduce the uncertainty since the biggest uncertainty comes from the back azimuth rather than the epicentral distance.

“The estimation of the marsquake hypocenter location leads to relatively small uncertainty along the longitude direction (from the uncertainty of the epicentral distance) for the position of the bounce point, but large uncertainty along the latitude direction (from the uncertainty of the back azimuth).”

We also mentioned why we can reduce the timing uncertainty. MQS’s pick is a preliminary value and needs

to be relatively large so that MQS does not miss any important signal that others later might claim to be the arrivals.

“Note that this relatively large uncertainty of ± 10 s was assigned in Horleston et al. (2022), since there are some energies within this relatively long window (i.e., the PP/SS phases and the precursors we found in this study). Consistently, our preferred PP/SS picks are also within this uncertainty range.”

(7) The only discussion I found on the effect of a possible mantle triplication phase is in the caption of Extended Data Fig. 3. Nothing is said about this in the text. The figure caption does not specify how the triplication operator was calculated, what the corresponding velocity model looks like, what it is based on – i.e. some independent information from InSight or predictions from equations of state and mineralogy, or just the desire to reproduce the raw data shown at the bottom right of Fig. 3 more closely. What is the evidence for a mantle discontinuity “at around 1000 km” depth? This whole point needs some more elaboration.

A39: Thank you for pointing it out. Previously, we tried to understand why there are two peaks for the PP precursor but only one peak for the main PP phase, and found that to reproduce the raw data more closely, we would have to introduce a mantle triplication phase (not based on mineralogy).

This time, as suggested by Reviewer #3, we also analyzed the data on the pressure sensor and found that the second peak for the PP precursor is very likely due to wind noise injection (Fig. S13). Therefore, this time, we filtered the data into the frequency band that is the least affected by the wind noise. The updated Fig. 2 no longer exhibits a second peak for the PP precursor.

(8) The inversion basically tries to fit two numbers within their uncertainty range, the differential arrival time of the phase and the relative amplitude. No waveforms are computed here, e.g. to visualize how well the computed results fit the actual data. This is a simple, but possible, way to invert the data. In sharp contrast to this, one of the most advanced modeling tools available is used to calculate the finite frequency effects for the size of the bounce point, SPEC-FEM3D_GLOBE, instead of e.g. looking at the Fresnel zone for SS and PP. How is this difference in approach warranted? What exactly are the benefits of using a 3D simulation here if the velocity model in the code is obviously 1D (i.e. it has a global Moho depth of 48 km, unlike the value proposed for the bounce point)? Why not run some full wavefield simulation at least for a preferred velocity model to see if it can reproduce the waveforms, if the tools are readily available?

A40: First, we changed the interface depth in SPEC-FEM3D_GLOBE from 48 km to 23 km (the value from this study) to re-calculate the sensitivity kernel. However, SPEC-FEM3D_GLOBE cannot simulate waveform at periods shorter than 4 s due to its computational cost. Instead, we chose another software, QSEIS (Wang 1999), to calculate the full 3-D wavefield using the 1-D model. A benchmark between the QSEIS synthetics and our modeling approach is shown in Fig. S11.

“For the synthetic waveform shown in Fig. 3, we extracted the source time function from the data after the polarization filtering (i.e., from -5.9 to 9.7 s for SS and from -3.2 to 5.5 s for PP). To simulate the precursor generated at each interface, we first multiplied the amplitude of the source time function with the calculated

synthetic amplitude ratio. Then, we shifted the source time function according to the differential travel time derived from equations (1) and (2). We have also compared this modeling approach with another software QSEIS (Wang 1999), which calculated the full wave field, and found consistent amplitude ratio when there is no significant interference from other phases. If there is interference (e.g., depth phases), since it shows up after both the main SS phase and its precursor, it does not significantly affect the phase identification (Fig. S11).”

As the reviewer suggested, we also included another inversion based on the waveform information.

“The inversions, in the main text, try to fit the arrival time and amplitude ratio measurements within their uncertainty range (i.e., one standard deviation). We have also performed another inversion, for the S-waves, based on the misfit (i.e., the cross-correlation coefficient) between the polarization-filtered data and the synthetic waveforms for comparisons (Fig. S12). The inverted models based on the waveform misfit do not significantly deviate from those from the arrival time and amplitude ratio measurements, though the inverted depth of the interface is about 5-km deeper. This slightly greater depth of the interface from the waveform misfit inversion is understandable since the left sidelobe of the main SS phase is also included. However, this pulse is not considered in the travel time measurements since this signal shows inconsistency between the raw and polarization-filtered waveforms. Therefore, we prefer the inversion approach, in the main text, where only the reliable signals are picked.”

(9) The authors state at several places in the text that the discontinuity at around 20 km depth they find is between the crust and mantle (i.e. l. 133, l. 138ff, where they say everything below 20 km is the mantle). Only in the abstract, they acknowledge that this could also be an intra-crustal discontinuity (l. 27). In the discussion, they point out the similarity between the discontinuity observed here and the discontinuity at about 20 km depth beneath the InSight lander (l. 170ff). However, the most recent publications point to this layer beneath the lander being within the crust, and not the crust-mantle interface (Duran et al. 2021, Kim et al. 2021, Wieczorek et al. 2022). If the Moho depth at the bounce point is in fact approximately 20 km, this would be another striking difference between the two locations, especially since gravity and topography predict similar crustal thickness assuming the same crustal density across the northern lowlands. This would mean that this assumption is fundamentally flawed, or the Moho depth at InSight is in fact 20 km. They authors need to discuss these implications, if they intend suggest a 20 km deep Moho – i.e. in how far does this modify the picture of the Martian crust derived in Wieczorek et al. (2022) in terms of origin and mineralogy of the crust, distribution of heat producing elements etc.?

A41: We agree with the reviewer that the three-layer crustal model (i.e., a thicker crust) was supported by ample seismological studies (e.g., Kim et al. (2021) and Durán et al. (2022)), and we prefer not to insist on the two-layer model. This time, we specifically named the discontinuity at ~ 23 km as ‘intra-crustal interface’ throughout the manuscript and in Fig. 1b to clarify this.

(10) The geological discussion in the paragraph starting at l. 194 needs some rephrasing: the first sentence sounds like impact craters are related to the dichotomy, which is probably not what is meant. L. 200 – what’s meant by “volcanic or sedimentary ash”? Doesn’t ash always require a volcanic process to create it? L. 201 – please rephrase “ a large impact cratering history” – is this a long history of impact cratering, a

history of large impact craters, a cratering history with a large impact? L. 202 – Impact cratering has been suggested as the reason/cause for low seismic velocities on the Moon. L. 208 – what is meant with Noachian rocks that are resurfaced by Hesperian to Amazonian volcanic rocks? Hesperian and Amazonian volcanics deposited on top of Noachian rocks? Please clarify here and in l. 218.

A42: Thank you for pointing these out. We have rephrased those sentences:

L. 194, L. 200, L. 201 removed (since we reduced the content on conclusion 2, but emphasized conclusion 1, please refer to #A5).

L. 202:

“On the Moon, impact cratering has also been suggested as the cause for low seismic velocities or crustal porosity (e.g., Wieczorek et al. 2013, Milbury et al. 2015, Soderblum et al. 2015).”

L. 208:

Yes, the Noachian rocks are covered by Hesperian to Amazonian volcanic rocks in some places. To better illustrate this, we added a geological map (Tanaka et al., 2014) near the bounce point (Fig. S8).

“This region is transected by the crustal dichotomy boundary and thus includes heavily impacted early- to mid-Noachian crustal rocks to the south (that in places are covered (resurfaced) by Hesperian to Amazonian volcanic rocks), Hesperian to Amazonian transitional units at the boundary, and Hesperian to late Amazonian volcanic rocks to the north.”

(11) Finally, I am wondering about the title – if “Second seismic anchor point” is not somewhat hard to make sense of for the general reader. Something like “Martian crustal structure distant from the InSight landing site” or “Differences in Martian crustal structure at a location distant from the InSight landing site” or similar might be more accessible.

A43: Thank you for the suggestion. We have changed the title:

“New Constraints on the Martian Crust Away From the InSight Landing Site”

Minor points:

L. 34 needs to be clarified – like “ indicating that this layer is either a local structure beneath the lander, or related to the type of geological unit, or it has been obscured at the bounce point by later processes.”

It’s also not clear what the difference between local structure beneath the lander and the type of geological unit is – in which other way could we define local structure on Mars than by looking at the geology?

A44: We have removed this sentence since it was previously related to conclusion 2 (please refer to #A5).

L. 53 should be “using P-to-s receiver functions”

A45: corrected.

Fig. 1c: Line 74 states that the epicenter is given by the dashed area, but there are two dashed areas in Fig. 1c – please be more specific. The figure caption also talks about a fan-shaped dashed area that is narrowed down to a shaded region close to Valles Marineris – I can only discern one darkly shaded region around the black star indicating the epicenter and am at a loss to identify any fan-shape. Please try to mark and distinguish all the shaded areas on the map better.

A46: We have updated the quality of the figure, please refer to the updated Fig. 1.

L. 185ff – This sentence is misleading. The authors state a possible SS precursor observed at -8.5 s is more likely a P-wave since it is seen dominantly on the vertical component, which would not be expected for SS precursors. The next part of the argument sounds like an incorrect back azimuth was somehow involved in transferring energy from the vertical (P-wave dominated) to the transverse (analyzed here for SS-precursors) component, which is not possible since the rotation for the backazimuth only involves radial and transverse components. I guess what's meant is spilling over of P-wave energy from the radial to the transverse component by either an incorrect backazimuth or anisotropy, but this has to be phrased more clearly – it is never indicated here that the radial component might also contain P-wave energy. Also, it would be helpful to show any figure of the radial component at all, e.g. include it in Extended Data Fig. 1, to get a full overview of the analyzed signal.

A47: Thank you for pointing it out. This time, we included the radial component in Fig. S2. We also changed the statement in the manuscript:

“However, on the tangential component, there are other signals (e.g., at -5.0 s and -8.5 s) (Fig. 2a) that are mainly visible in the raw data but much attenuated after the polarization filtering. We propose such pulses are not likely to be the SS precursors because there are strong energies on the vertical component at the same arrival time (Fig. S2a), which should not be expected for the shear waves.”

However, this time we prefer not to mention the possible sources for this suspicious energy since we do not have enough evidence for them. Instead, we performed another inversion treating this suspicious energy as a seismic signal and found that this will not affect the location of the interface at ~ 20 km.

“Nevertheless, we performed another inversion assuming two crustal layers since if either of these two signals were to be another seismic precursor, another shallow interface beneath the bounce point would be required to generate this phase. Inversion results (Fig. 4b) show that adding an extra discontinuity in the top 12 km does not significantly change the location of the second layer (24.0 ± 4.4 km, in Fig. 4e) and the S-wave speeds within it (2.4 ± 0.4 km/s, in Fig. 4f).”

L. 325f: Reference to Fig. 1 is incorrect – there are no seismic waveforms shown in Fig. 1.

A48: Corrected, it should be Fig. 2.

Reviewer #3 (Remarks to the Author):

This paper describes the identification of SS and PP precursors generated from the Martian crust-mantle discontinuity at a distance of 4000km from the InSight landing site using the most distant marsquake recorded by SEIS, S0976a. The authors further invert the observed data and obtain a crustal interface at a depth of 20 ± 5 km, in agreement with results from previous studies. However, a shallower interface at a depth of 8 ± 2 km previously reported at the InSight landing site by various studies was not observed at this location. The paper is well written, clear in its presentation, and certainly adds to our understanding of the Martian crustal structure. The techniques utilised are simple and elegant and primarily utilise polarization filtering of three component ground motion recordings along with forward and inverse modelling (i.e., grid-search approach). The authors then present their interpretation and reasoning for the discrepancy between the resulting structure at the bounce point and the structure at the landing site. With that said, the paper could be improved after taking into account the following concerns.

1) The authors show the raw and polarization filtered data for both the tangential and the vertical displacements in Figure 2. While the SS precursor phase is more evident in the tangential component, the PP precursor seen in the filtered vertical component appears less evident due to its seemingly lower amplitude ratio and possible phase distortion. It would be helpful to report this amplitude ratio for the PP precursor and the PP phase (and those derived from receiver function models) in the main text, as has been done for the SS precursor.

A49: This time, we measured the amplitude ratio for the PP precursor in different frequency bands on both the raw data and polarization-filtered data (Fig. S3). We also compared the data and predictions (from the receiver function models) in Fig. S5, and mentioned it in the main manuscript:

“The amplitude ratio between the PP precursor and the PP phase is about $43 \pm 8\%$ (Fig. S3), which is also comparable to the synthetic amplitude ratio (i.e., about 25-40% in Fig. S5).”

2) The possible second positive pulse in the PP precursor has been attributed to mantle triplication. This is presented quite clearly in Figure S3. However, it is not clear how the authors derive the mantle triplication operator without any knowledge of the deeper structure. The nature of the seismic discontinuity at 1000 km depth referred to here seems rather vague. Is it just an assumption, a possible inference, or has this been established by previous studies? Some more clarity about this is needed.

A50: It was an assumption. This time, we have found a better and more robust explanation for the second positive pulse in the PP precursor, i.e., wind injection. Please refer to #A39.

3) Figure 3 (a) shows polarisation filtered data for the tangential component, while Figure 3 (b) shows the raw displacement data for the vertical component. It would be preferable to use the polarization filtered data in both cases. There also seems to be no scale on the y axis in all the subplots. It would be a good idea to add it to have amplitude comparisons available.

A51: This time, to be more consistent, we show the polarization filtered data (in red) for both PP and SS waves in Fig. 3. In addition, we plotted the raw data in light grey color for comparison. We also added the y axis in Fig. 3. We note that the y axis in Fig. 2 is the true amplitude (m), however the y axis in Fig. 3 is the normalized amplitude since the polarization filtering weights each data point, therefore, there is no true amplitude for the polarization-filtered data.

4) In the Discussion section, the authors note that the polarization filter can suppress low amplitude signals that are more strongly affected by environmental noise. The authors are also aware of the contaminations observed in the seismic data recorded by SEIS and confirm that there are no glitches around the observed PP and SS phases in So976a. However, the SEIS data also commonly shows strong environmental injections, as confirmed using data from the APSS (e.g., Charalambous et al., 2021). Have such correlations been taken into consideration by this study? The observations, if contaminated, can mimic phases that otherwise might not exist or show wrong polarization/amplitude properties that could lead to wrong conclusions (e.g., about the shallower discontinuity).

A52: Thank you for this comment. The possible contamination from environmental noise (i.e., wind injection) should be considered. This time, we analyzed the wind noise from both the pressure sensor and the weather-sensitive lander mode and wrote a section ‘comodulation analysis’ in the method part. Based on this new analysis, we employed the seismic data in the frequency band that is the least affected by the wind injection.

Please also refer to #A39 and Fig. S13.

5) Although the analysis is impressive, these results have been based on just a single, small magnitude event. Terrestrial data has often shown that clear detection and identification of PP and SS precursors can be quite challenging even when a large number of events are available (e.g., Lessing et al., 2015). Stacking methods are therefore often employed. How do the authors explain this? A small discussion about this would certainly be helpful.

A53: The challenges mentioned in Lessing et al. (2015) is that most of the PP precursors on Earth show very small reflection coefficients, i.e., less than 5%. The reflection coefficient is also a function of incidence angle, and sometimes, the incidence angle of the PP ray paths yields a smaller coefficient than the SS precursor). But the most important reason why stacking is indeed needed on Earth is the relatively small impedance contrast.

We reproduced the amplitude ratio between the SS precursor and the main SS phase following Lessing et al. (2015) for the 410-km and 660-km on Earth, and the crustal discontinuities on Mars. The significant velocity contrast across the crustal interfaces on Mars leads to an average amplitude ratio of ~ 30%, i.e., 6 to 10 times larger than the one across the mantle transition zone discontinuities on Earth. We mention this in the manuscript:

“Synthetic waveforms show that such precursors are observable for a single event, without the need for stacking. Waveform stacking is a technique widely used on Earth to detect the mantle transition zone

discontinuities (e.g., Flanagan and Shearer, 1998; Lessing et al., 2015). Contrary to those Earth studies, for which stacking is needed to enhance the signal, the analysis presented here does not require stacking owing to the relatively high velocity (or impedance) contrast across the intra-crustal interface on Mars (Fig. S4).”

Reviewer #1 (Remarks to the Author):

The authors have addressed a lot of my comments and the data and processing sections have considerably improved. However, I still find room for improvement. For this purpose, I am attaching an annotated manuscript with suggestions, comments, and questions that I would like the authors to carefully consider. Apart from missing references, illegible figure labels, typos, etc., I would particularly appreciate a comparison of the inverted models with more recent results. There is little point in comparing with initial results only. Finally, what I found very unsatisfactory is the final discussion, which is simply too qualitative and needs to be considerably improved and strengthened. In summary, I recommend moderate revisions.

In their review of the second version of this manuscript, reviewer #1 added some comments to the manuscript file. These comments were forwarded to the authors, who replied as included in this Peer Review File

Reviewer #2 (Remarks to the Author):

In response to the reviews, the authors made a big effort to revise the paper by describing the data processing more clearly and thoroughly, making the figures more accessible, and consolidating their conclusions by additional analyses (e.g. the whole section on Comodulation Analysis in the Methods is new, the benchmarking of forward modeling approaches in Supplementary Figure 11, the modeling of the SS precursors with two crustal layer, and the inversion based on waveform misfit). In my opinion, this really improved the manuscript. They also updated references to include the most recent publications (e.g. Huang et al. 2022, Kim et al. 2022, Posiolova et al., 2022 and Wang and Tkalcic 2022). In doing so, they addressed all of my comments and questions and the paper is basically fit for publication – it provides new and important knowledge on the crust of Mars.

However, I have two minor comments on parts newly added to the manuscript that can (and should) be addressed very quickly and would clarify some points:

(1) Fig. 2 (b) includes the positive parts of the autocorrelation coefficients in gray shading, which is a good idea. It should be stated that the coefficients for the tangential trace (top) are multiplied by (-1), though, to fit into the plot. Also, I was wondering if the scale for those traces is the same as given on the Y-axis – the maximum of the autocorrelation coefficients at zero time should be 1 then, which it does not seem to be.

(2) The authors state several times (e.g. l. 359, 575) that they are using 10,000 models from the receiver function study (Knapmeyer-Endrun et al., 2021). It would be good if they could be more specific about which models they are using – there were two different inversion approaches in that study, each applied to a high-frequency and a low-frequency data set, and each resulting in 5,000 models for each dataset. So in total, that would make 20,000 models with three crustal layers. Which ones of those are used here?

Reviewer #3 (Remarks to the Author):

This paper describes the identification of SS and PP precursors generated from the Martian crust-mantle the discontinuity at a distance of 4000km from the InSight landing site using the most distant marsquake recorded by SEIS, S0976a. The authors further invert the observed data and obtain a crustal interface at a depth of 20 ± 5 km, in agreement with results from previous studies. However, a shallower interface at a

depth of 8 ± 2 km previously reported at the InSight landing site by various studies was not observed at this location. The paper is well written, clear in its presentation, and certainly adds to our understanding of the Martian crustal structure. The techniques utilised are simple and elegant and primarily utilise polarization filtering of three component ground motion recordings along with forward and inverse modelling (i.e., grid-search approach). The authors then present their interpretation and reasoning for the discrepancy between the resulting structure at the bounce point and the structure at the landing site.

The revised manuscript has addressed all my major concerns using good arguments and explanations. The review process has certainly led to an improvement in the quality of the manuscript, and I now find it suitable for publication. I will therefore recommend the editor to publish this manuscript.

Thank you for your time and expertise with our manuscript. We appreciate the constructive feedback from the three reviewers. Below, we outline the reviewers' comments (**in blue**), and respond to each point-by-point (**in black**). New changes or additions to the manuscript are shown **in red**.

REVIEWER COMMENTS

Reviewer #1 (Remarks to the Author):

The authors have addressed a lot of my comments and the data and processing sections have considerably improved. However, I still find room for improvement. For this purpose, I am attaching an annotated manuscript with suggestions, comments, and questions that I would like the authors to carefully consider. Apart from missing references, illegible figure labels, typos, etc., I would particularly appreciate a comparison of the inverted models with more recent results. There is little point in comparing with initial results only. Finally, what I found very unsatisfactory is the final discussion, which is simply too qualitative and needs to be considerably improved and strengthened. In summary, I recommend moderate revisions.

A2: Thanks for the annotated manuscript, we have carefully addressed the comments, and refined the reference list, figure labels and typos.

We also added an extra comparison with more recent results, i.e., Durán et al. (2022). Since the interface depth beneath the landing site in Durán et al. (2022) is consistent with the discontinuity we found at the bounce point, we found our conclusion is still valid. We want to note that we did not follow Reviewer #1's suggestion to add the entire velocity profiles in Durán et al. (2022) into Fig. 4 (it will make the current Fig. 4 too busy). Instead, we simply plotted the interface depth in Durán et al. (2022) as a horizontal bar in Fig.

4e.

For the final discussion, we added more contents to strengthen the discussion and emphasize the significance of this study. In addition, we performed a calculation to quantitatively connect the seismic velocity jumps to the porosity changes. We added a new Methods section, a new Supplementary Figure 16, and a Supplementary Table 1:

Methods section: Porosity Effects

For the origin of the '20-km' seismic interface identified both beneath the landing site and the bounce point, a variety of mostly non-mutually exclusive factors could play a role. Our currently favored hypothesis is the removal of pore space at depth. To quantitatively assess the influence of porosity, we assume that the velocity jump across the '20-km' interface is affected by porosity alone.

Above the '20-km' interface, the average P- and S-wave velocities beneath the InSight landing site are 4.1 km s⁻¹ and 2.3 km s⁻¹, respectively¹¹. At the bounce point, the average P- and S-wave velocities above the interface are 3.8 km s⁻¹ and 2.4 km s⁻¹, respectively (i.e., Fig. 4). Below the interface, the average P- and S-wave velocities beneath the InSight landing site are 5.5 km s⁻¹ and 3.1 km s⁻¹, respectively¹¹. At the bounce point, we prefer to use the velocities inferred from the landing site, since the precursors have fewer constraints on the structures below the discontinuity. This choice should be reasonable because Supplementary Fig. 12 indicates that there are no significant differences in the lower crust between those two places.

Then, we follow the scattering theory⁵⁴ to estimate both the P- and the S-wave velocity of porous basalt^{55,56} as a function of porosity (Supplementary Fig. 16). In the calculation, a pore aspect ratio of 0.1 was assumed³¹, and the pore space can be filled with gas (i.e., carbon dioxide) or liquid water (see Supplementary Table 1 for properties of the materials). We did not consider water ice as the pore-filling medium because temperatures are too high to freeze water at this depth range, near the equator^{57,58}.

If the pore spaces are removed below the '20-km' seismic interface, we should expect a corresponding porosity close to 0%. However, Supplementary Figs. 16a and 16c respectively show that an S-wave velocity of 3.1 km s⁻¹ corresponds to a porosity of ~10%, and a P-wave velocity of 5.5 km s⁻¹ corresponds to a porosity of ~8%. We note that this discrepancy is likely due to the assumption that the solid groundmass is 100% basalt in the calculation. Consistently, Wiczorek et al. (2022) (ref.³⁴) found that the density for known igneous lithologies can vary by 25% relatively and also proposed that the Martian crust could be more felsic than typical basaltic materials. Therefore, we followed Kilburn et al. (2022) (ref.⁵⁸) to consider the other felsic end-member igneous rock, i.e., plagioclase feldspar⁵⁹. Supplementary Figs. 16b and 16d show that, when considering plagioclase to be the groundmass, the corresponding porosities are only ~ 0% and ~ 5%, inferred from the S- and P-wave velocities, respectively.

Supplementary Figs. 16b and 16d also show that an S-wave velocity of 2.3 – 2.4 km s⁻¹ corresponds to a porosity of ~16%, and a P-wave velocity of 3.8 - 4.1 km s⁻¹ corresponds to a porosity of ~15%. This indicates that the velocity difference above and below the pore closure depth can be explained by a porosity reduction of 10 - 16%, assuming an aspect ratio of 0.1.

Supplementary Figure 16. Analysis of the porosity effects on the seismic wave velocities.

- (a) S-wave speed of porous basalt with intrusions of carbon dioxide (in red) and liquid water (in blue) as a function of porosity (with the aspect ratio of 0.1). The solid black line represents the seismic velocity above the '20-km' discontinuity beneath the lander site¹. The dashed black and grey lines indicate the seismic wave velocities above the pore closure depth at the lander site and the bounce point, respectively. The corresponding porosity estimations are also indicated (see Methods section).
- (b) Similar to (a), but for plagioclase feldspar as the groundmass.
- (c) Similar to (a), but for the P-wave velocity.
- (d) Similar to (b), but for the P-wave velocity.

Supplementary Table 1. Elastic properties of the materials in the porosity simulation.

Density, bulk modulus, and shear modulus for the solid groundmass of the basalt, plagioclase, and the two pore fluids. Properties of basalt were taken from Christensen (1972) (ref.⁹) and Heap (2019) (ref.¹⁰). Properties of plagioclase were taken from Woeber et al. (1963) (ref.¹¹). Properties of liquid water and CO₂ (for a pressure of 0.1 MPa and a temperature of 20 °C) were taken from Heap (2019) (ref.¹⁰).

Table S1 Elastic properties of the materials in the porosity simulation

Material	Density (kg/m ³)	Bulk modulus (GPa)	Shear modulus (GPa)	References
Basalt	2,900	80.0	40.0	Christensen (1972); Heap (2019)
Plagioclase	2,630	75.6	25.6	Woeber et al. (1963)
Liquid water (20 °C)	1,000	2.2	0	Heap (2019)
CO ₂ (20 °C)	1.8	0.0001	0	Heap (2019)

Reviewer #2 (Remarks to the Author):

In response to the reviews, the authors made a big effort to revise the paper by describing the data processing more clearly and thoroughly, making the figures more accessible, and consolidating their conclusions by additional analyses (e.g. the whole section on Comodulation Analysis in the Methods is new, the benchmarking of forward modeling approaches in Supplementary Figure 11, the modeling of the SS precursors with two crustal layer, and the inversion based on waveform misfit). In my opinion, this really improved the manuscript. They also updated references to include the most recent publications (e.g. Huang et al. 2022, Kim et al. 2022, Posiolova et al., 2022 and Wang and Tkalcic 2022). In doing so, they addressed all of my comments and questions and the paper is basically fit for publication – it provides new and important knowledge on the crust of Mars.

However, I have two minor comments on parts newly added to the manuscript that can (and should) be addressed very quickly and would clarify some points:

- (1) Fig. 2 (b) includes the positive parts of the autocorrelation coefficients in gray shading, which is a good idea. It should be stated that the coefficients for the tangential trace (top) are multiplied by (-1), though, to fit into the plot. Also, I was wondering if the scale for those traces is the same as given on the Y-axis – the maximum of the autocorrelation coefficients at zero time should be 1 then, which it does not seem to be.

A3: Thanks for pointing it out. This time, we clarified in the caption that the top trace is multiplied by -1: **“The positive parts of the auto-correlation coefficients for the vertical and tangential components are shown as shaded waveforms on the top (multiplied by -1 to fit into the plot) and bottom axes, respectively.”**

We have also rescaled the autocorrelation traces to make the maximum value at zero time to be 1.

(2) The authors state several times (e.g. l. 359, 575) that they are using 10,000 models from the receiver function study (Knapmeyer-Endrun et al., 2021). It would be good if they could be more specific about which models they are using – there were two different inversion approaches in that study, each applied to a high-frequency and a low-frequency data set, and each resulting in 5,000 models for each dataset. So in total, that would make 20,000 models with three crustal layers. Which ones of those are used here?

A4: Previously, we used 10,000 models (out of 20,000 models) from Knapmeyer-Endrun et al. (2021). We agree with the reviewer that if we only included parts of the models, we should specify which models are used.

This time, we included all the 20,000 models from Knapmeyer-Endrun et al. (2021) and updated Fig. 4.

Reviewer #3 (Remarks to the Author):

This paper describes the identification of SS and PP precursors generated from the Martian crust-mantle the discontinuity at a distance of 4000km from the InSight landing site using the most distant marsquake recorded by SEIS, S0976a. The authors further invert the observed data and obtain a crustal interface at a depth of 20 ± 5 km, in agreement with results from previous studies. However, a shallower interface at a depth of 8 ± 2 km previously reported at the InSight landing site by various studies was not observed at this location. The paper is well written, clear in its presentation, and certainly adds to our understanding of the Martian crustal structure. The techniques utilised are simple and elegant and primarily utilise polarization filtering of three component ground motion recordings along with forward and inverse modelling (i.e., grid-search approach). The authors then present their interpretation and reasoning for the discrepancy between the resulting structure at the bounce point and the structure at the landing site.

The revised manuscript has addressed all my major concerns using good arguments and explanations. The review process has certainly led to an improvement in the quality of the manuscript, and I now find it suitable for publication. I will therefore recommend the editor to publish this manuscript.

A5: Thank you and we appreciate the review process.